# Trade-offs in modeling context dependency in complex trait genetics

Eric Weine[1,2,3], Samuel Pattillo Smith[1,2], Rebecca Kathryn Knowlton[4], Arbel Harpak[1,2]*

[1]Department of Integrative Biology, The University of Texas at Austin, Austin, United States; [2]Department of Population Health, The University of Texas at Austin, Austin, United States; [3]Department of Human Genetics, University of Chicago, Chicago, United States; [4]Department of Statistics and Data Sciences, The University of Texas at Austin, Austin, United States

## eLife Assessment

It is known from model organisms that genes' effects on traits are often modulated by environmental variables, but similar gene-by-environment (GxE) interactions have been difficult to detect using statistical analyses of genomic data, e.g., in humans. This study introduces a new framework to estimate gene-by-environment effects, treating it as a bias-variance tradeoff problem. The authors **convincingly** show that greater statistical power can be achieved in detecting GxE if an underlying model of polygenic GxE is assumed. This polygenic amplification model is a truly novel view with **fundamental** promise for the detection of GxE in genomic datasets, especially with continued development to detect more complex signals of amplification.

*For correspondence:
arbelharpak@utexas.edu

**Abstract** Genetic effects on complex traits may depend on context, such as age, sex, environmental exposures, or social settings. However, it remains often unclear if the extent of context dependency, or gene-by-environment interaction (GxE), merits more involved models than the additive model typically used to analyze data from genome-wide association studies (GWAS). Here, we suggest considering the utility of GxE models in GWAS as a trade-off between bias and variance parameters. In particular, we derive a decision rule for choosing between competing models for the estimation of allelic effects. The rule weighs the increased estimation noise when context is considered against the potential bias when context dependency is ignored. In the empirical example of GxSex in human physiology, the increased noise of context-specific estimation often outweighs the bias reduction, rendering GxE models less useful when variants are considered independently. However, for complex traits, we argue that the joint consideration of context dependency across many variants mitigates both noise and bias. As a result, polygenic GxE models can improve both estimation and trait prediction. Finally, we exemplify (using GxDiet effects on longevity in fruit flies) how analyses based on independently ascertained 'top hits' alone can be misleading, and that considering polygenic patterns of GxE can improve interpretation.

## Introduction

In organisms and study systems where the environment can be tractably manipulated, gene-by-environment interactions (GxE) are the rule, not the exception (**El Soda et al., 2014**; **Vieira et al., 2000**; **Des Marais et al., 2013**; **Smith and Kruglyak, 2008**; **Paaby and Rockman, 2014**). Yet, in complex (polygenic) human traits, there are but a few cases in which models that incorporate GxE explain data—such as genome-wide association study (GWAS) data—better than parsimonious models

that assume additive contributions of genetic and environmental factors (*Munafò et al., 2014*; *Kraft and Aschard, 2015*; *Sella and Barton, 2019*). This is true for both physical environments but also for other definitions of 'E', broadly construed to be any context that modifies genetic effects, such as age, sex, or social setting (*Zhu et al., 2023*; *Schwaba et al., 2023*; *Elgart et al., 2022*; *Duncan and Keller, 2011*; *Gibson and Lacek, 2020*; *Brown et al., 2016*; *Ge et al., 2017*; *Balliu et al., 2021*). GWAS commonly estimate marginal additive effects of an allele on a trait. The estimand here can be thought of as the average effect of the allele over a distribution of multidimensional contexts (*Veller et al., 2023*). With this view, some differences in allelic effects across contexts are likely omnipresent, but may very well be small, such that the cost of including additional parameters (for context-specific effects) outweighs the benefit of measuring heterogeneous effects. Here, we consider this problem and its connection to the currently underwhelming utility of GxE models in GWAS. First, we rigorously describe the statistical trade-off involved in estimating context specificity at the level of a single variant. Then, we highlight ways in which this trade-off might change as we consider GxE in complex traits, involving numerous genetic variants simultaneously. We begin by framing the problem of estimating context specificity at an individual variant as a bias-variance trade-off. For example, consider the estimation of an allelic effect on lung cancer risk that depends on smoking status. When the allelic effect is estimated from a sample without considering smoking status, the estimate would be biased with respect to the true effect in smokers. We can estimate the effect separately in smokers and non-smokers to eliminate the bias, but the consideration of the additional parameters—smoking status-specific effects—has an associated cost of increasing the estimation variance, compared to an estimator that ignores smoking status. This bias-variance trade-off is closely related to the 'signal-to-noise' ratio, where the signal of interest is the true difference in context-specific allelic effects. To demonstrate this trade-off in real data, we consider sex-specific effects on physiological traits in humans. We show that for the majority of traits, it is typically unhelpful to model sex dependency for individual sites since the increase in noise vastly outweighs the signal. We then consider the extension to GxE in complex traits. Complex trait variation is primarily due to numerous genetic variants of small effects distributed throughout the genome (*Fisher, 1930*; *Falconer and Mackay, 1996*; *Yengo et al., 2022*; *Zwick et al., 2000*). Simultaneously considering GxE across multiple variants may decrease estimation noise if the extent and mode of context specificity is similar across numerous variants. This would tilt the scale in favor of context-dependent estimation. In addition, we show how conventional approaches for detecting and characterizing GxE, which focus on the most significant associations, may lead to erroneous conclusions. Finally, we discuss implications for complex trait prediction (with polygenic scores). We suggest a future focus on prediction methods that empirically learn the extent and nature of context dependency by simultaneously considering GxE across many variants.

## Results and discussion
### Modeling context-dependent effect estimation as a bias-variance trade-off
### The problem setup

We consider a sample of $n + m$ individuals characterized as being in one of two contexts, $A$ or $B$ of the individuals are in context $A$ with the remaining $m$ individuals in context $B$. We measure a continuous trait for each individual, denoted by

$$\overbrace{y_1, \ldots, y_n}^{A}, \overbrace{y_{n+1}, \ldots, y_{n+m}}^{B}.$$

We begin by considering the estimation of the effect of a single variant on the continuous trait. We assume a generative model of the form

$$y_i \sim \begin{cases} \mathcal{N}(\alpha_A + \beta_A g_i, \sigma_A^2) & \text{if } i \in \{1, \ldots, n\} \\ \mathcal{N}(\alpha_B + \beta_B g_i, \sigma_B^2) & \text{if } i \in \{n+1, \ldots, n+m\}, \end{cases} \tag{1}$$

where $\beta_A$ and $\beta_B$ are fixed, context-specific effects of a reference allele at a biallelic, autosomal variant $i$, $g_i \in \{0, 1, 2\}$ is the observed reference allele count. $\alpha_A$ and $\alpha_B$ are the context-specific intercepts,

corresponding to the mean trait for individuals with zero reference alleles in context $A$ and $B$, respectively. $\sigma_A^2$ and $\sigma_B^2$ are context-specific observation variances. We would like to estimate the allelic effects $\beta_A$ and $\beta_B$.

## Estimation approaches

We compare two approaches to this estimation problem. The first approach, which we refer to as GxE estimation, is to stratify the sample by context and separately perform an ordinary least squares (OLS) regression in each sample. This approach yields two estimates, $\hat{\beta}_A$ and $\hat{\beta}_B$, the OLS estimates of $\beta_A$ and $\beta_B$ of the generative model in *Equation 1*, respectively. This estimation model is equivalent to a linear model with a term for the interaction between context and reference allele count, in the sense that context-specific allelic effect estimators have the same maximum likelihood estimators in the two models (see Appendix 1). The second approach, which we refer to as additive estimation, is to perform an OLS regression on the entire sample and use the allelic effect estimate to estimate both $\beta_A$ and $\beta_B$. We denote this estimator as $\hat{\beta}_{A\cup B}$, to emphasize that the regression is run on all individuals from context $A$ and context $B$. This estimation model posits that for $i = 1, \ldots, n + m$,

$$y_i \sim \mathcal{N}(\alpha_{A\cup B} + \beta_{A\cup B}g_i, \sigma_{A\cup B}^2), \tag{2}$$

where $\alpha_{A\cup B}$ is the mean trait value for an individual with zero reference alleles, $\beta_{A\cup B}$ is the additive allelic effect and $\sigma_{A\cup B}^2$ is the observation variance which is independent of context. Notably, this model differs from the generative model assumed above: $\beta_{A\cup B}$ may not equal $\beta_A$ and $\beta_B$; in addition, this model ignores heteroskedasticity across contexts.

## Error analysis

We focus on the mean squared error (MSE) of the additive and GxE estimators for the allelic effect in context $A$. The estimator minimizing the MSE may differ between contexts $A$ and $B$, but the analysis for context $B$ is analogous. When selecting between these two estimation approaches, a bias-variance decomposition of the MSE is useful. Based on OLS theory (*Casella and Berger, 2021*, Theorem 11.3.3), under the model specified above, we have

$$\hat{\beta}_A \sim \mathcal{N}(\beta_A, V_A),$$

where $V_A = \dfrac{\sigma_A^2}{\sum_{i=1}^{n}(g_i - \bar{g}_A)^2}$ and $\bar{g}_A$ is the mean genotype of individuals in context $A$. The unbiasedness of the GxE estimator implies

$$MSE(\hat{\beta}_A, \beta_A) = V_A,$$

where $MSE(\hat{\beta}_A, \beta_A)$ is the mean squared error of estimating $\beta_A$ with $\hat{\beta}_A$. The case of the additive estimator, $\hat{\beta}_{A\cup B}$, is a bit more involved. As we show in the Methods section, we can write

$$\hat{\beta}_{A\cup B} = \omega_A \hat{\beta}_A + \omega_B \hat{\beta}_B \tag{3}$$

for non-negative weights $\omega_A$ and $\omega_B$ (that need not sum to 1). Further, we show in *Equation 7* of the Methods section that $\omega_A \propto nH_A$ and $\omega_B \propto mH_B$, where $H_A$ and $H_B$ are the sample heterozygosities in contexts $A$ and $B$, respectively. Using *Equation 3*, we may write

$$MSE(\hat{\beta}_{A\cup B}, \beta_A) = Bias^2(\hat{\beta}_{A\cup B}, \beta_A) + Var(\hat{\beta}_{A\cup B})$$
$$= \left((\omega_A - 1)\beta_A + \omega_B\beta_B\right)^2 + \omega_A^2 V_A + \omega_B^2 V_B,$$

where $V_B$ is defined analogously to $V_A$. Thus, with MSE as our metric for comparison, we prefer the GxE estimator in context $A$ when

$$MSE(\hat{\beta}_{A\cup B}, \beta_A) > MSE(\hat{\beta}_A, \beta_A),$$

or, if and only if

$$\left((\omega_A - 1)\beta_A + \omega_B\beta_B\right)^2 + \omega_A^2 V_A + \omega_B^2 V_B > V_A. \tag{4}$$

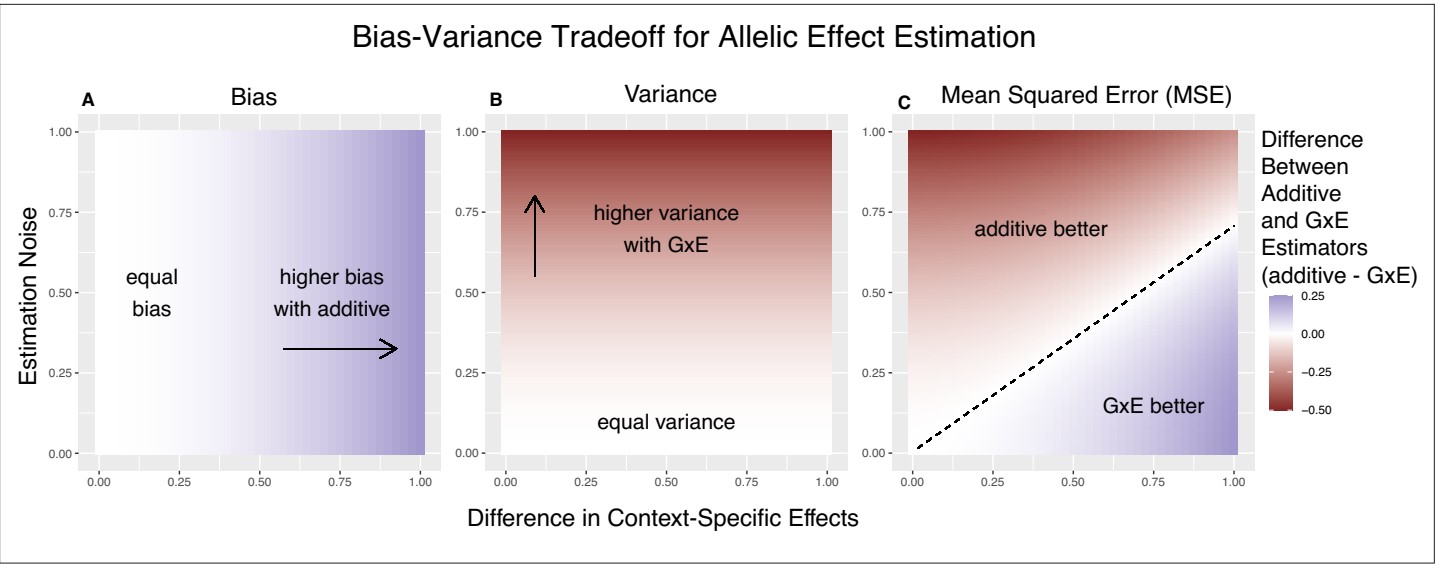

**Figure 1.** Bias-variance trade-off for single-site estimation with equal estimation noise and equal heterozygosity across contexts. The *x*-axis shows the difference in context-specific effects, while the *y*-axis shows the standard deviation of the context-specific estimators—both in raw measurement units. The color on the plot indicates the difference between the additive and gene-by-environment interaction (GxE) estimators in bias (**A**), variance (**B**), or mean squared error (MSE) (**C**). (**A**) Only the additive estimator is potentially biased. The bias is proportional to the difference in context-specific effects and independent of the estimation noise. (**B**) The difference in variance is proportional to context-specific estimation noise and independent of the difference of context-specific effects. (**C**) The decision boundary is linear in both the estimation noise and the difference between context-specific effects.

We refer to *Equation 4* as the 'decision rule', since it guides us on the more accurate estimator; to minimize the MSE, we will use the context-specific estimator if and only if the inequality is satisfied. To gain some intuition about the important parameters here, we first consider the case of equal allele frequencies (and hence equal heterozygosities) in both contexts and equal estimation variance in both contexts. In this case, the GxE estimator is advantaged by larger context specificity (larger $|\beta_A - \beta_B|$) and disadvantaged by larger estimation noise (larger $V_A = V_B$) (*Figure 1*). In fact, the decision boundary (i.e. the point at which the two models have equal MSE) can be written as a linear combination of $|\beta_A - \beta_B|$ and $\sqrt{V_A}$ (*Figure 1C*). In this special case, we show in the Methods section that *Equation 4* is an equality when

$$\sqrt{\frac{m}{2n}}|\beta_A - \beta_B| - \sqrt{V_A} = 0. \tag{5}$$

More generally, in the case where $H_A = H_B$ but $V_A \neq V_B$, we show in the Methods section that we can write *Equation 4* as

$$\frac{(\beta_A - \beta_B)^2}{V_A} > \frac{1 + \omega_A}{1 - \omega_A} - \frac{V_B}{V_A}. \tag{6}$$

This dimensionless re-parameterization of the decision rule makes explicit its dependence on three factors. $\frac{(\beta_A - \beta_B)^2}{V_A}$ can be viewed as the 'signal-to-noise' ratio: it captures the degree of context specificity (the signal) relative to the estimation noise in the focal context, $A$. $\frac{1 + \omega_A}{1 - \omega_A}$ is the relative contribution to heterozygosity, which equals the relative contribution to variance in the independent variable of the OLS regression of *Equation 2*. $\frac{V_B}{V_A}$ is the ratio of context-specific estimation noises. In Appendix 1, we extend the decision rule for the case of a continuous context variable. For a given trait and context, we can consider the behavior of the decision rule across variants with variable allele frequencies and allelic effects. The ratio of estimation noises, $r := \frac{V_A}{V_B}$, will not be constant. However, in some cases, considering a fixed $r$ across variants is a good approximation. In GWAS of complex traits, each variant often explains a small fraction of trait variance. As a result, the estimation noise is effectively a matter of trait variance and heterozygosity alone. If per-site heterozygosity is similar in strata $A$ and $B$, as it is, for example, for autosomal variants in biological males and females, $r$ is approximately fixed

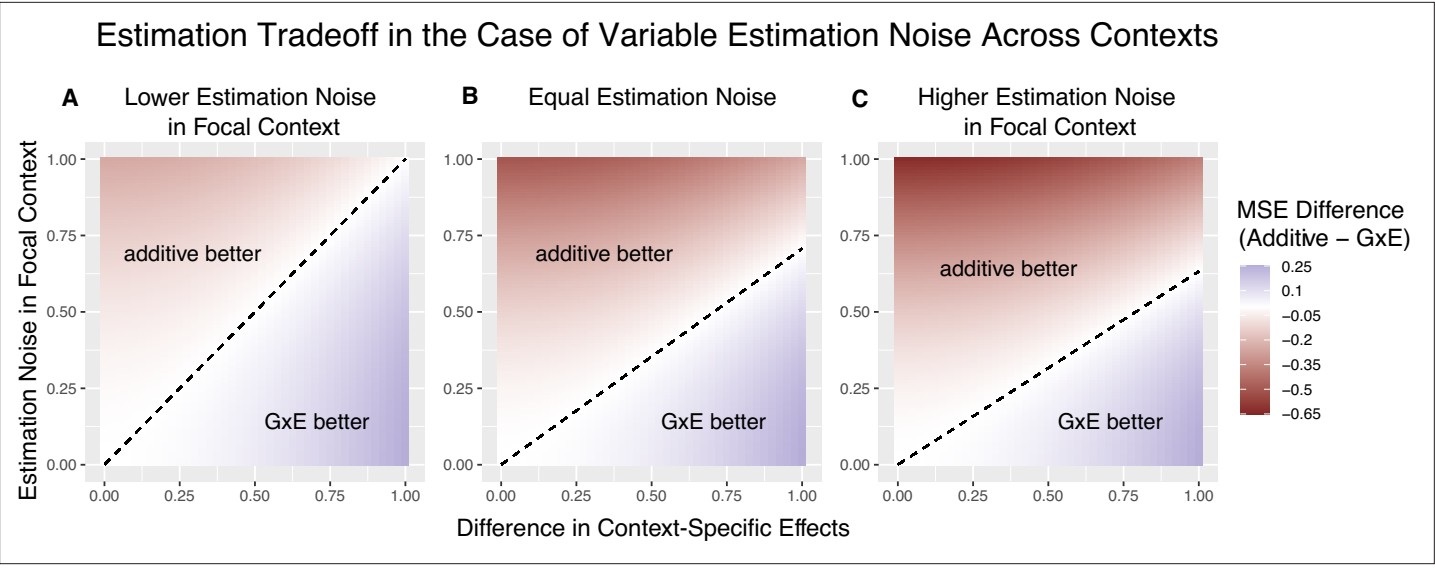

**Figure 2.** The decision boundary with different ratios of context-specific estimation noises. In all panels, the heterozygosity of the variant is assumed to be equal across contexts. The $x$ and $y$ axes are the same as in **Figure 1**. (**A**) Estimation noise in the focal context, $A$, is half that of the other context, $B$. (**B**) Estimation noise is equal in both contexts. (**C**) Estimation noise in focal context is double that of the other context.

across variants (**Zhu et al., 2023**). **Figure 2** illustrates the linearity of the decision boundary under the assumption that $r$ is fixed across variants. It also shows that the slope of the decision boundary changes as a function of $r$. Intuitively, we are less likely to prefer GxE estimation for the noisier context. In fact, for sufficiently small values of $r$ (e.g. $r < \frac{1}{3}$ for $\omega_A = \frac{1}{2}$), $\frac{1+\omega_A}{1-\omega_A} - \frac{V_B}{V_A}$ will be negative. This corresponds to the situation where $V_A \ll V_B$, in which case the additive estimator is never preferable to the GxE estimator in estimating $\beta_A$, as the signal-to-noise ratio is always non-negative. Typically, this will also imply that the additive estimator is greatly preferable for estimating $\beta_B$, as $\hat{\beta}_B$ will be extremely noisy.

It is natural to ask where the decision rule of **Equation 4** falls with respect to empirical GWAS data. We considered the example of biological sex as the context (GxSex), and examined sex-stratified GWAS data across 27 continuous physiological traits in the UK Biobank (**Bycroft et al., 2018**; **Zhu et al., 2023**). For each of 9 million variants, we estimated the difference in sex-specific effects and the variance of each marginal effect estimator in males. Then, using an estimate of the ratio of sex-specific trait variances as a proxy for the ratio of estimation variances of males and females, we approximated the linear decision boundary between the additive and GxE estimators (**Figure 3A and B**; **Appendix 1—figure 2**, **Appendix 1—figure 3**). To demonstrate the accuracy of our decision rule, we employed a data-splitting technique where we estimate the MSE difference between estimators in a training set and evaluate the accuracy in a holdout set (**Appendix 1—figure 1**). For almost all traits examined, very few allelic effects in males are expected to be more accurately estimated using the male-specific estimator (usually between 0% and 0.1%). Notable exceptions to this rule are testosterone, sex hormone binding globulin (SHBG), and waist-to-hip ratio adjusted for body mass index, for which roughly 0.5% of allelic effects are expected to be better estimated with the GxE model (**Figure 3B**). However, when considering only SNPs that are genome-wide significant in males (marginal p-value $< 5 \times 10^{-8}$ in males), many traits show a much larger proportion of effects that would be better estimated by the GxE model. At an extreme, for testosterone, all genome-wide significant SNPs are expected to be better estimated by the GxE model. In addition, a large fraction of genome-wide significant effects are better estimated with the GxE model for creatinine (62%), arm fat-free mass (24%), waist-to-hip ratio (19%), and SHBG (18%) as well (**Figure 3D**).

The decision rule we derived could potentially guide more accurate allelic effect estimation approaches. However, the consideration of GxE pattern sharing across many variants (polygenic GxE) can alter both bias and variance and therefore the trade-off. In our discussion of complex traits that follows, we therefore expand on the rule through qualitative consequences of polygenic GxE, and no longer stick to the analytical single variant rule.

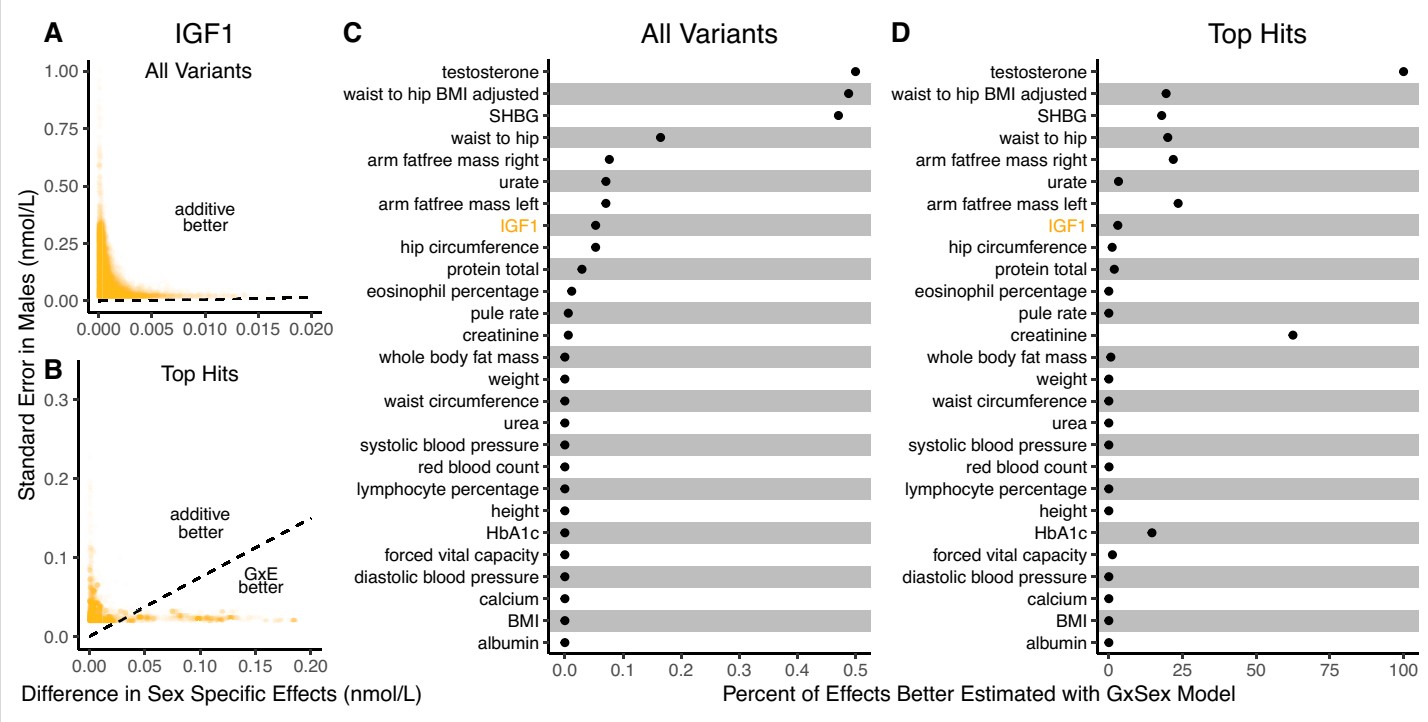

**Figure 3.** Applying the decision rule to sex-dependent effects on human physiological traits. (**A, B**) The x-axis shows the estimated absolute difference between the effect of variants in males and females. The y-axis shows the measured standard error for each variant in males, the focal context here. The dashed line shows the decision boundary for effect estimation in males. The difference in mean squared error (MSE) between estimation methods increases linearly with distance from the dashed line, as in *Figure 2*. If a variant falls above (below) the line, the additive (gene-by-environment interaction [GxE]) estimator has a lower MSE. (**A**) shows a random sample of 15K single nucleotide variants whereas (**B**) shows only variants with a marginal p-value less than $5 \times 10^{-8}$ in males. (**C, D**) The percent of effects in males which would be better estimated by the GxE estimator, across continuous physiological traits. (Note the difference in scale between the two panels.) To estimate these percentages, one single nucleotide variant is sampled from each of 1700 approximately independent autosomal linkage blocks, and this procedure is repeated 10 times. Shown are average percentages across the 10 iterations.

## Context dependency in complex traits

At the single variant level, and specifically when variants are considered independently from one another, we have discussed how the accurate estimation of allelic effects can be boiled down to a bias-variance trade-off. For complex traits, genetic variance is often dominated by the contribution of numerous variants of small effects that are best understood when analyzed jointly (*Sella and Barton, 2019*; *Sinnott-Armstrong et al., 2021*; *Shi et al., 2016*; *Boyle et al., 2017*; *Liu et al., 2019*; *Wray et al., 2018*; *Yengo et al., 2022*). It stands to reason that to evaluate context dependence in complex traits, we would also want to jointly consider polygenic patterns, rather than just the patterns at the loci most strongly associated with a trait (*Urbut et al., 2019*; *Gibson and Lacek, 2020*; *Zhang et al., 2021*; *Paaby and Gibson, 2016*; *Aschard et al., 2017*; *Des Marais et al., 2013*). Motivated by this rationale, we recently inferred polygenic GxSex patterns in human physiology (*Zhu et al., 2023*). One pattern that emerged as a common mode of GxSex across complex physiological traits is 'amplification': a systematic difference in the magnitude of genetic effects between the sexes. Moving beyond sex and considering any context, amplification can happen if, for example, many variants regulate a shared pathway that is moderated by a factor—and that factor varies in its distribution among contexts. Amplification is but one possible mode of polygenic GxE, but can serve as a guiding example for ways in which GxE may be pervasive but difficult to characterize with existing approaches (*Zhu et al., 2023*; *Gibson and Dworkin, 2004*; *Miao et al., 2022*; *Balliu et al., 2021*). In what follows, we will therefore use the example of pervasive amplification (across causal effects) to illustrate the interpretive advantage of considering context dependency across variants jointly, rather than independently.

## A focus on 'top hits' may lead to mischaracterization of polygenic GxE

A common approach to the analysis of context dependency involves two steps. First, categorization of context dependency (or lack thereof) is performed for each variant independently. Second, variants falling under each category are counted and annotated across the genome. Some recent examples of this approach toward the characterization of GxE in complex traits include studies of GxSex effects on flight performance in *Drosophila* (*Spierer et al., 2021*), GxSex effects on various traits in humans (*Traglia et al., 2022*; *Bernabeu et al., 2021*), and GxDietxAge effects on body weight in mice (*Wright et al., 2022*). Characterizing polygenic trends by summarizing many independent hypothesis tests may miss GxE signals that are subtle and statistically undetectable at each individual variant, yet pervasive and substantial cumulatively across the genome. To characterize polygenic GxE based on just the 'top hits' may lead to ascertainment biases, with respect to both the pervasiveness and the mode of GxE across the genome. Much like the heritability of complex traits is thought to be due to the contribution of many small (typically sub-significant) effects (*Boyle et al., 2017*; *Sinnott-Armstrong et al., 2021*), when GxE is pervasive we may expect that the sum of many small differences in context-specific effects accounts for the majority of GxE variation. For concreteness, we consider in more depth one recent study characterizing GxDiet effects on longevity in *Drosophila melanogaster* (*Pallares et al., 2023*). In this study, Pallares et al. tracked caged fly populations given one of two diets: a 'control' diet and a 'high-sugar' diet. Across 271K single nucleotide variants, the authors tested for association between alleles and their survival to a sampling point (thought of as a proxy for 'lifespan' or 'longevity') under each diet independently. Then, they classified variants according to whether or not their associations with survivorship were significant under each diet as follows:

1. Significant under neither diet → classify as *no effect*.
2. Significant when fed the high-sugar diet, but not when fed the control diet → classify as *high-sugar-specific effect*.
3. Significant when fed the control diet, but not when fed the high-sugar diet → classify as *control-specific effect*.
4. Significant under both diets → classify as *shared effect*.

This authors' choice of four categories a variant may fall into may be motivated by the wish to test for the presence of 'cryptic genetic variation'—genetic variation that is maintained in a context where it is functionally neutral but carries large effects in a new or stressful context (*Gibson and Dworkin, 2004*; *Paaby and Rockman, 2014*; *Young et al., 2016*; *Des Marais et al., 2013*). Indeed, of the variants Pallares et al. classified as having an effect (one-hundredth of variants tested), approximately 31% were high-sugar specific, while the remaining 69% of the variants were shared. Fewer than 1% were labeled as having control-specific effects. They concluded that high-sugar-specific effects on longevity are pervasive, compatible with the hypothesis of widespread cryptic genetic variation for longevity. This characterization of GxE, based on 'top hits', places an emphasis on the context(s) in which trait associations are statistically significant, rather than on estimating how the context-specific effects covary. In addition, this particular classification system also does not cover all possible ways in which context-specific effects may differ. In Appendix 1, we discuss these interpretation difficulties further.

We next show that a generative model that differs qualitatively from the cryptic genetic variation model yields results that are highly similar to those observed by Pallares et al. We simulated data under pervasive amplification. Specifically, we sampled from a mixture of 40% of variants having no effect under either diet and 60% of variants having an effect under both diets—but exactly 1.4× larger under a high-sugar diet. We then simulated the noisy estimation of these effects and employed the classification approach of Pallares et al. to the simulated data (Methods). The patterns of allelic effects in the control compared to high-sugar contexts were qualitatively similar in the experimental data and our pervasive amplification simulation. This is true both genome-wide (*Figure 4A* compared to *Figure 4B*) and for the set of variants classified as significant with their classification approach (*Figure 4C* compared to *Figure 4D*). The similarity of ascertained variants further highlights caveats of interpretation based on the classification of 'top hits': despite the fact that we did not simulate any variants that only have an effect under the high-sugar diet, approximately 36% of significant variants were classified as specific to the high-sugar diet (green points in *Figure 4D*), comparable to the 31% of variants classified as high-sugar specific in the experimental data (*Figure 4C*). These variants simply have sub-significant associations in the control group and significant associations in the high-sugar group. In addition, every variant in the shared category (blue points in *Figure 4D*) in fact has a larger

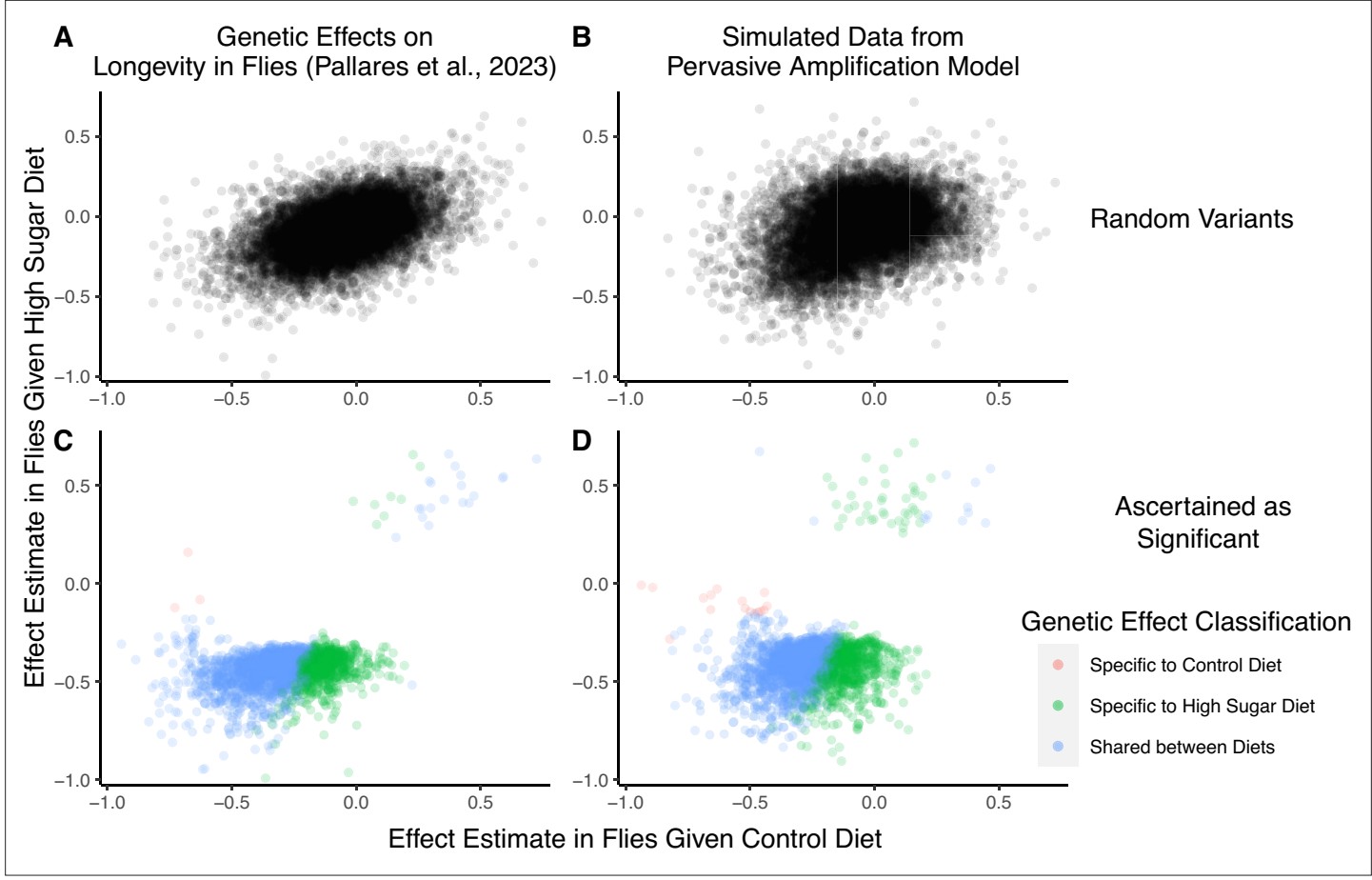

**Figure 4.** A focus on top hits may be lead to mischaracterization of polygenic gene-by-environment interactions (GxE). (**A**) Data from an experiment measuring allelic effects on longevity in caged flies given one of two diets, 'control' and 'high sugar'. Shown are allelic effect estimates under each diet for a random sample of approximately 12K variants. (**B**) Simulated data where all true allelic effects are exactly 1.4 times larger under a high-sugar diet. The effects are estimated with sampling noise mimicking the Pallares et al. data. (**C**) Allelic effect estimates of variants ascertained as significant and classified as 'diet-specific' or 'shared' by Pallares et al. (**D**) Simulated effects ascertained as significant and classified using a similar procedure to that applied in (**C**). While the generative mode of GxE we used in our simulations was not considered by Pallares et al., the simulation results (left panels) closely match the patterns observed in their data (right panels) across all effects (top panels) and as reflected via their classification approach (bottom panels).

effect in the high-sugar diet than in the control diet, which cannot be captured by the classification system itself but represents the only mode of GxE in our simulation.

To recap, we simulated a mode of GxE that is not considered in Pallares et al. (i.e. pervasive amplification) and that is at odds with their conclusions about evidence for a large discrete class of SNPs with diet-specific effects (i.e. cryptic genetic variation). The close match of our simulation to the empirical results of Pallares et al. therefore illustrates that the characterization of GxE via hypothesis testing and classification at each variant independently may lead to erroneous interpretation when applied to empirical complex trait data as well. In Appendix 1, we show that a reanalysis of the Pallares et al. data that is based on estimating the covariance of allelic effects is directly consistent with pervasive amplification as well (***Appendix 1—figure 4***). In conclusion, the classification of 'top hits' alone may not be representative of the extent of GxE nor of the most pervasive modes of GxE.

## The utility of modeling GxE for complex trait prediction

Modeling context dependency of genetic effects may hold the potential for constructing polygenic scores that are more accurate or improve their portability across contexts (***Patel et al., 2022***; ***Miao et al., 2022***; ***Turley et al., 2018***; ***Spence et al., 2022***; ***Wang et al., 2024***; ***Smith et al., 2025***). Evidence for the utility of GxE models in polygenic score prediction, however, has been underwhelming and GxE models are still

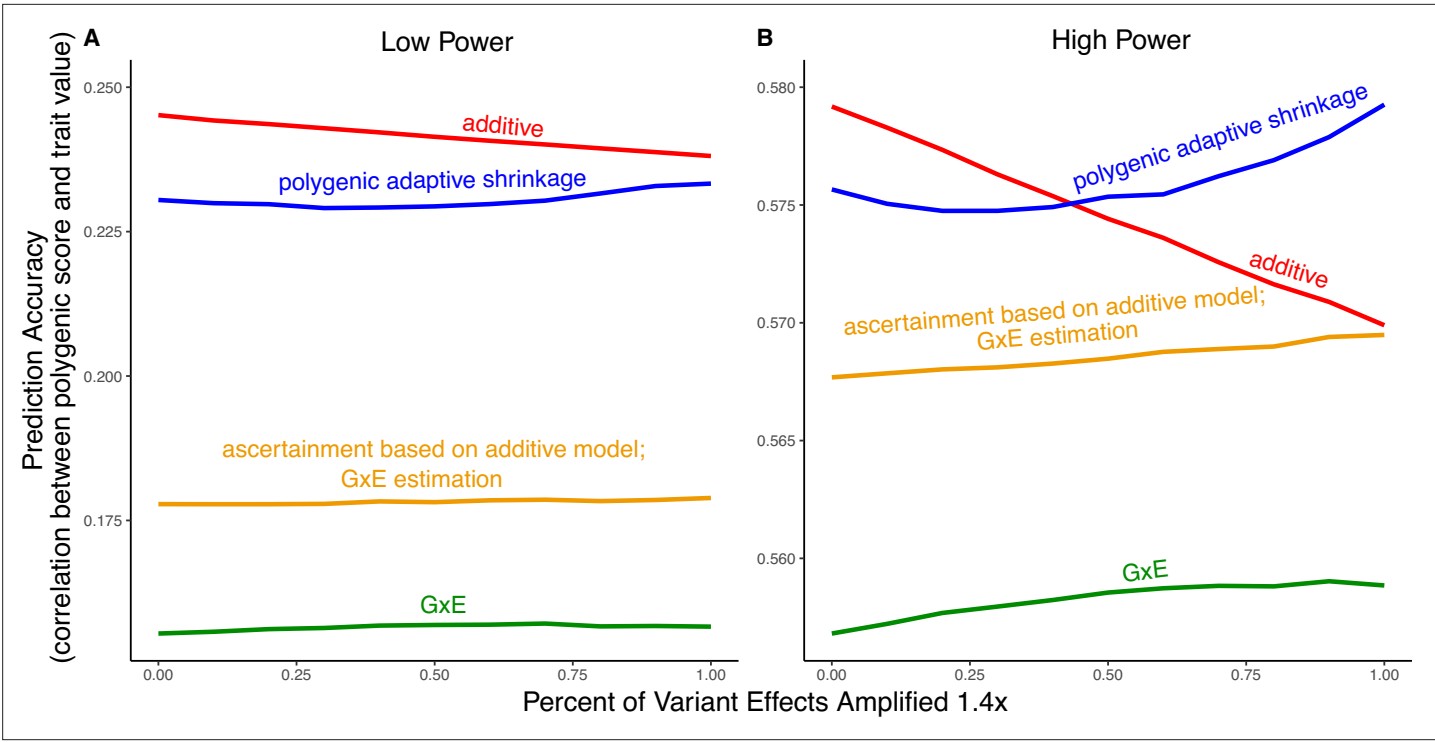

**Figure 5.** Polygenic score performance for context-dependent prediction models. In each simulation, a genome-wide association study (GWAS) is performed on 5000 biallelic variants, half of which have no effect in either context. Of the other half, some percent of the variants (indicated on the *x*-axis) had effects 1.4× larger in one of contexts and the remaining SNPs had equal effects in both contexts. The broad sense heritability was set to $0.4$ in all simulations. The *y*-axis shows the average, over $11,000$ simulations, of the out-of-sample Pearson correlation between polygenic score and trait value. (**A**) Results with a GWAS sample size of 1000 individuals. (**B**) Results with a GWAS size of $50,000$ individuals.

rarely applied (*Zhu et al., 2023*; *Schwaba et al., 2023*). A key reason behind this apparent discrepancy is the bias-variance trade-off for individual variants discussed above. If context-specific effects are similar—a likely possibility for highly polygenic traits with the majority of heritability owing to small causal effects—then additive models will tend to outperform (*Fisher, 1930*; *Falconer and Mackay, 1996*; *Hill et al., 2008*; *Young, 2019*). This is because the unbiasedness of GxE estimation does not make up for the cost of additional estimator variance, resulting from sample stratification by context or the addition of explicit interaction terms (*Schwaba et al., 2023*). We exemplify the relative importance of variance compared to bias in polygenic scoring using simulations. We continue with the generative model of pervasive amplification as an example. Namely, we simulated a GWAS of a continuous trait with independent effects in 2500 variants (50% of variants included in the GWAS). Effects were either the same in two contexts, *A* and *B*, or 1.4 times larger in context *B*. The GWAS is conducted with either a small sample size or a large sample size, conferring low or high statistical power, respectively. We then constructed polygenic scores using 833 variants (corresponding to one-third of the causal variants), which were ascertained as most significantly associated with the trait according to either the additive model (orange and red in *Figure 5*) in or context-specific hypothesis tests (green and blue in *Figure 5*).

Even in settings with pervasive GxE, additive polygenic scores (red lines in *Figure 5*) outperformed context-specific scores (green lines in *Figure 5*). The advantage of the additive model is manifested in two ways: more accurate estimation, as discussed above, but also better identification of true associations with the trait. We considered the two advantages separately. It is sometimes better to ascertain variants using the lower variance approach and estimate effects using the lower-bias approach. In our simulations, this strategy (orange lines in *Figure 5*) was preferable to using the GxE model for both ascertainment and estimation (green line). It was not preferable to using the additive model (red line) for both approaches, but it was the preferable strategy under a slightly different parametric regime, corresponding to more GxE (*Appendix 1—figure 5*). Finally, we considered a polygenic GxE approach, as implemented in 'multivariate adaptive shrinkage' (*mash*) (*Urbut et al., 2019*), a method to estimate context-specific effects by leveraging common patterns of effect covariance between contexts

observed across the genome. *mash* models the underlying distribution of effects in all contexts as a mixture of zero-centered multivariate normal distributions with different covariance structures (as well as the null matrix, to induce additional shrinkage). After estimating this distribution via maximum likelihood, *mash* uses it as a prior to obtain posterior effect estimates for each variant in each context. As a result, posterior effect estimates across contexts regress toward commonly observed patterns of covariance of allelic effects across contexts. In our simulations, in the presence of substantial amplification, the polygenic adaptive shrinkage approach outperformed all other methods as long as the study was adequately powered (*Figure 5B*). This is thanks to the unique ability (compared to the three other approaches) to leverage the sharing of signals across variants, including the extent and nature of context dependency. With low power, however, the additive model performed best (*Figure 5A*). We attribute this to the variance cost associated with the polygenic adaptive shrinkage approach—driven by the estimation of additional parameters for capturing the genome-wide covariance relationships.

## Conclusion

When genetic variants are considered independently, the estimation of their effects in different contexts can be boiled down to a bias-variance trade-off. For complex traits, we show through example that further considering polygenic patterns of GxE can be key for understanding context-dependent genetic architecture and to aid in prediction. The notion that complex trait analyses should combine observations at top associated loci alongside polygenic trends has gained traction with additive models of trait variation; it may be similarly important in our understanding of context dependency.

## Methods
### Expressing the additive estimator as a linear combination of GxE estimators

In this section, we prove the result of *Equation 3*, stating that

$$\hat{\beta}_{A\cup B} = \omega_A \hat{\beta}_A + \omega_B \hat{\beta}_B$$

for some non-negative weights $\omega_A$ and $\omega_B$. To do this, we will need some additional notation. Let $\bar{g}_A$ denote the average number of effect alleles in individuals in context $A$, and let $\bar{g}_{A\cup B}$ denote the average effect allele count across all individuals. Similarly, let $\bar{y}_A$ denote the average trait value in context $A$, and let $\bar{y}_{A\cup B}$ denote the average trait value across all individuals. As an OLS estimator, the context-specific estimator is defined as

$$\hat{\beta}_A = \frac{\sum_{i=1}^{n}(g_i - \bar{g}_A)(y_i - \bar{y}_A)}{\sum_{i=1}^{n}(g_i - \bar{g}_A)^2}$$

$$= \frac{\sum_{i=1}^{n}(g_i - \bar{g}_A)y_i - \sum_{i=1}^{n}(g_i - \bar{g}_A)\bar{y}_A}{\sum_{i=1}^{n}(g_i - \bar{g}_A)^2}$$

$$= \frac{\sum_{i=1}^{n}(g_i - \bar{g}_A)y_i - \bar{y}_A\sum_{i=1}^{n}(g_i - \bar{g}_A)}{\sum_{i=1}^{n}(g_i - \bar{g}_A)^2}$$

$$= \frac{\sum_{i=1}^{n}(g_i - \bar{g}_A)y_i}{\sum_{i=1}^{n}(g_i - \bar{g}_A)^2}, \text{ since } \sum_{i=1}^{n}(g_i - \bar{g}_A) = 0.$$

Similarly, the additive estimator can be written as

$$
\hat{\beta}_{A\cup B} = \frac{\sum_{i=1}^{n+m}(g_i - \bar{g}_{A\cup B})(y_i - \bar{y}_{A\cup B})}{\sum_{i=1}^{n+m}(g_i - \bar{g}_{A\cup B})^2}
$$

$$
= \frac{\sum_{i=1}^{n+m}(g_i - \bar{g}_{A\cup B})y_i}{\sum_{i=1}^{n+m}(g_i - \bar{g}_{A\cup B})^2} \quad \text{(by the same logic as above)}
$$

$$
= \frac{\sum_{i=1}^{n}(g_i - \bar{g}_{A\cup B})y_i + \sum_{i=n+1}^{n+m}(g_i - \bar{g}_{A\cup B})y_i}{\sum_{i=1}^{n+m}(g_i - \bar{g}_{A\cup B})^2}.
$$

We will show that the weights in *Equation 3* depend on the effect allele frequency in the two contexts, $f_A$ and $f_B$. We will assume mean-centered traits, such that $\sum_{i=1}^{n} y_i = 0$ and $\sum_{i=n+1}^{n+m} y_i = 0$. We note that mean-centering is inconsequential for effect estimation. We can then write

$$
\hat{\beta}_{A\cup B} = \frac{\sum_{i=1}^{n}(g_i - \bar{g}_{A\cup B})y_i + \sum_{i=n+1}^{n+m}(g_i - \bar{g}_{A\cup B})y_i}{\sum_{i=1}^{n+m}(g_i - \bar{g}_{A\cup B})^2}
$$

$$
= \frac{\sum_{i=1}^{n}(g_i - \bar{g}_{A\cup B})y_i}{\sum_{i=1}^{n+m}(g_i - \bar{g}_{A\cup B})^2} + \frac{\sum_{i=n+1}^{n+m}(g_i - \bar{g}_{A\cup B})y_i}{\sum_{i=1}^{n+m}(g_i - \bar{g}_{A\cup B})^2}
$$

$$
= \frac{\sum_{i=1}^{n}(g_i - \bar{g}_A + (\bar{g}_A - \bar{g}_{A\cup B}))y_i}{\sum_{i=1}^{n+m}(g_i - \bar{g}_{A\cup B})^2} + \frac{\sum_{i=n+1}^{n+m}(g_i - \bar{g}_B + (\bar{g}_B - \bar{g}_{A\cup B}))y_i}{\sum_{i=1}^{n+m}(g_i - \bar{g}_{A\cup B})^2}
$$

$$
= \frac{\sum_{i=1}^{n}(g_i - \bar{g}_A)y_i + \sum_{i=1}^{n}(\bar{g}_A - \bar{g}_{A\cup B})y_i}{\sum_{i=1}^{n+m}(g_i - \bar{g}_{A\cup B})^2} + \frac{\sum_{i=n+1}^{n+m}(g_i - \bar{g}_B)y_i + \sum_{i=n+1}^{n+m}(\bar{g}_B - \bar{g}_{A\cup B})y_i}{\sum_{i=1}^{n+m}(g_i - \bar{g}_{A\cup B})^2}
$$

$$
= \frac{\sum_{i=1}^{n}(g_i - \bar{g}_A)y_i + (\bar{g}_A - \bar{g}_{A\cup B})\sum_{i=1}^{n} y_i}{\sum_{i=1}^{n+m}(g_i - \bar{g}_{A\cup B})^2} + \frac{\sum_{i=n+1}^{n+m}(g_i - \bar{g}_B)y_i + (\bar{g}_B - \bar{g}_{A\cup B})\sum_{i=n+1}^{n+m} y_i}{\sum_{i=1}^{n+m}(g_i - \bar{g}_{A\cup B})^2}
$$

$$
= \frac{\sum_{i=1}^{n}(g_i - \bar{g}_A)y_i}{\sum_{i=1}^{n+m}(g_i - \bar{g}_{A\cup B})^2} + \frac{\sum_{i=n+1}^{n+m}(g_i - \bar{g}_B)y_i}{\sum_{i=1}^{n+m}(g_i - \bar{g}_{A\cup B})^2} \quad \text{(by our assumption of mean centered traits)}
$$

$$
= \frac{\sum_{i=1}^{n}(g_i - \bar{g}_A)y_i}{\sum_{i=1}^{n+m}(g_i - \bar{g}_{A\cup B})^2} \cdot \frac{\sum_{i=1}^{n}(g_i - \bar{g}_A)^2}{\sum_{i=1}^{n}(g_i - \bar{g}_A)^2} + \frac{\sum_{i=n+1}^{n+m}(g_i - \bar{g}_B)y_i}{\sum_{i=1}^{n+m}(g_i - \bar{g}_{A\cup B})^2} \cdot \frac{\sum_{i=n+1}^{n+m}(g_i - \bar{g}_B)^2}{\sum_{i=n+1}^{n+m}(g_i - \bar{g}_B)^2}
$$

$$
= \frac{\sum_{i=1}^{n}(g_i - \bar{g}_A)y_i}{\sum_{i=1}^{n}(g_i - \bar{g}_A)^2} \cdot \frac{\sum_{i=1}^{n}(g_i - \bar{g}_A)^2}{\sum_{i=1}^{n+m}(g_i - \bar{g}_{A\cup B})^2} + \frac{\sum_{i=n+1}^{n+m}(g_i - \bar{g}_B)y_i}{\sum_{i=n+1}^{n+m}(g_i - \bar{g}_B)^2} \cdot \frac{\sum_{i=n+1}^{n+m}(g_i - \bar{g}_B)^2}{\sum_{i=1}^{n+m}(g_i - \bar{g}_{A\cup B})^2}
$$

$$
= \frac{\sum_{i=1}^{n}(g_i - \bar{g}_A)^2}{\sum_{i=1}^{n+m}(g_i - \bar{g}_{A\cup B})^2}\hat{\beta}_A + \frac{\sum_{i=n+1}^{n+m}(g_i - \bar{g}_B)^2}{\sum_{i=1}^{n+m}(g_i - \bar{g}_{A\cup B})^2}\hat{\beta}_B.
$$

Thus, $\omega_A = \dfrac{\sum_{i=1}^{n}(g_i - \bar{g}_A)^2}{\sum_{i=1}^{n+m}(g_i - \bar{g}_{A \cup B})^2}$ and $\omega_B = \dfrac{\sum_{i=n+1}^{n+m}(g_i - \bar{g}_B)^2}{\sum_{i=1}^{n+m}(g_i - \bar{g}_{A \cup B})^2}$ in *Equation 3*. We note that the numerator of $\omega_A$ is $n$ times the sample heterozygosity in context $A$, and the numerator of $\omega_B$ is $m$ times the sample heterozygosity in context $B$. Thus, we have shown that

$$\omega_A \propto nH_A \text{ and } \omega_B \propto mH_B, \tag{7}$$

where $H_A$ and $H_B$ are the sample heterozygosities in context $A$ and $B$, respectively. And, in the special case where $f_A = f_B$, because this implies that the sample heterozygosities will be approximately equal across contexts, we have that

$$\omega_A \propto nH_A \text{ and } \omega_B \propto mH_B. \tag{8}$$

## Linearity of the decision rule

In *Equation 5*, under the assumption that $V_A = V_B$ and $H_A = H_B$, the decision boundary is expressed as a linear function of $|\beta_A - \beta_B|$ and $\sqrt{V_A}$ as

$$\sqrt{\frac{m}{2n}}|\beta_A - \beta_B| > \sqrt{V_A}.$$

Here, we prove that the linearity of the decision rule holds in the more general case where $\dfrac{V_A}{V_B} = r$ for some fixed value of $r$. *Equation 5* then follows as a special case of this fact when $r = 1$. Starting from *Equation 4*, we prefer the GxE estimator to the additive estimator when estimating $\beta_A$ if

$$V_A < \omega_A^2 V_A + \omega_B^2 V_B + ((\omega_A - 1)\beta_A + \omega_B\beta_B)^2$$

$$\iff V_A < \omega_A^2 V_A + \frac{\omega_B^2}{r} V_A + ((\omega_A - 1)\beta_A + \omega_B\beta_B)^2$$

$$\iff V_A - \omega_A^2 V_A - \frac{\omega_B^2}{r} V_A < ((\omega_A - 1)\beta_A + \omega_B\beta_B)^2$$

$$\iff (1 - \omega_A^2 - \frac{\omega_B^2}{r})V_A < ((\omega_A - 1)\beta_A + \omega_B\beta_B)^2$$

$$\iff V_A < \frac{((\omega_A - 1)\beta_A + \omega_B\beta_B)^2}{1 - \omega_A^2 - \frac{\omega_B^2}{r}} \quad (\text{assuming } 1 - \omega_A^2 - \frac{\omega_B^2}{r} > 0)$$

$$\iff \sqrt{V_A} < \frac{|(\omega_A - 1)\beta_A + \omega_B\beta_B|}{\sqrt{1 - \omega_A^2 - \frac{\omega_B^2}{r}}} \quad (\text{again assuming } 1 - \omega_A^2 - \frac{\omega_B^2}{r} > 0)$$

If our assumption that $1 - \omega_A^2 - \dfrac{\omega_B^2}{r} > 0$ does not hold, we note that the GxE model is always preferable and technically speaking there exists no decision rule between the two models. Now, when heterozygosities (and thus minor allele frequencies) are equal across contexts, then *Equation 8* implies $\omega_A + \omega_B = 1$. Therefore, we may write the decision rule as

$$\sqrt{V_A} < \frac{|(1 - \omega_B - 1)\beta_A + \omega_B\beta_B|}{\sqrt{1 - \omega_A^2 - \frac{\omega_B^2}{r}}}$$

$$\iff \sqrt{V_A} < \frac{|\omega_B(\beta_B - \beta_A)|}{\sqrt{1 - \omega_A^2 - \frac{\omega_B^2}{r}}}$$

$$\iff \sqrt{V_A} < \frac{\omega_B}{\sqrt{1 - \omega_A^2 - \frac{\omega_B^2}{r}}}|\beta_A - \beta_B| \quad (\text{by properties of the absolute value})$$

$$\iff \sqrt{V_A} < \frac{1 - \omega_A}{\sqrt{1 - \omega_A^2 - \frac{(1 - \omega_A)^2}{r}}}|\beta_A - \beta_B|.$$

Here, we see that for any fixed $r$ the decision rule is linear with a slope determined by $r$ (**Figure 2**). Now, in the special case where $r = 1$, we have

$$\sqrt{V_A} < \frac{1 - \omega_A}{\sqrt{1 - \omega_A^2 - (1 - \omega_A)^2}} |\beta_A - \beta_B|$$

$$\iff \sqrt{V_A} < \frac{1 - \omega_A}{\sqrt{1 - \omega_A^2 - 1 - \omega_A^2 + 2\omega_A}} |\beta_A - \beta_B|$$

$$\iff \sqrt{V_A} < \frac{1 - \omega_A}{\sqrt{2\omega_A(1 - \omega_A)}} |\beta_A - \beta_B|$$

$$\iff \sqrt{V_A} < \sqrt{\frac{1 - \omega_A}{2\omega_A}} |\beta_A - \beta_B|$$

Now, substituting the definitions of $\omega_A$ and $\omega_B$ in the case of equal minor allele frequencies given in **Equation 8**, we can write

$$\sqrt{V_A} < \sqrt{\frac{1}{2}} \sqrt{\frac{1 - \frac{n}{n + m}}{\frac{n}{n + m}}} |\beta_A - \beta_B|$$

$$\iff \sqrt{V_A} < \sqrt{\frac{1}{2}} \sqrt{\frac{\frac{n + m}{m}}{\frac{n + m}{n}}} |\beta_A - \beta_B|$$

$$\iff \sqrt{V_A} < \sqrt{\frac{m}{2n}} |\beta_A - \beta_B|.$$

This inequality is instead an equality under the conditions stated in **Equation 5**. Finally, again using the definition of $\omega_A$ and $\omega_B$ given in **Equation 8**, we note that our assumption that $1 - \omega_A^2 - \frac{\omega_B^2}{r} > 0$ will always hold in the case of equal minor allele frequencies and $r = 1$, as

$$1 - \omega_A^2 - \frac{\omega_B^2}{r} = 1 - \frac{n^2}{(n + m)^2} - \frac{m^2}{(n + m)^2}$$

$$= \frac{(n + m)^2 - n^2 - m^2}{(n + m)^2}$$

$$= \frac{2nm}{(n + m)^2},$$

which is strictly positive.

## Re-parameterized decision rule in terms of unitless quantities

In **Equation 6**, under the assumption that $H_A = H_B$, we re-state the decision rule in terms of the signal-to-noise ratio. Here, we prove this result. From **Equation 4**, we have that we should select the GxE model to estimate $\beta_A$ if and only if

$$V_A < \omega_A^2 V_A + \omega_B^2 V_B + \left((\omega_A - 1)\beta_A + \omega_B \beta_B\right)^2$$

$$\iff 1 < \omega_A^2 + \omega_B^2 \frac{V_B}{V_A} + \frac{\left((\omega_A - 1)\beta_A + \omega_B \beta_B\right)^2}{V_A}$$

$$\iff 1 - \omega_A^2 < \omega_B^2 \frac{V_B}{V_A} + \frac{\left((\omega_A - 1)\beta_A + \omega_B \beta_B\right)^2}{V_A}.$$

Now, because $H_A = H_B$, we know by **Equation 8** that $\omega_A + \omega_B = 1$. Then, we may write the decision rule as

$$1 - \omega_A^2 < \omega_B^2 \frac{V_B}{V_A} + \frac{\left((1 - \omega_B - 1)\beta_A + \omega_B\beta_B\right)^2}{V_A}$$

$$\iff 1 - \omega_A^2 < \omega_B^2 \frac{V_B}{V_A} + \frac{\left(\omega_B(\beta_B - \beta_A)\right)^2}{V_A}$$

$$\iff 1 - \omega_A^2 < \omega_B^2 \frac{V_B}{V_A} + \omega_B^2 \frac{(\beta_A - \beta_B)^2}{V_A}$$

$$\iff \frac{1 - \omega_A^2}{\omega_B^2} < \frac{V_B}{V_A} + \frac{(\beta_A - \beta_B)^2}{V_A}$$

$$\iff \frac{1 - \omega_A^2}{(1 - \omega_A)^2} < \frac{V_B}{V_A} + \frac{(\beta_A - \beta_B)^2}{V_A}$$

$$\iff \frac{1 - \omega_A^2}{(1 - \omega_A)^2} - \frac{V_B}{V_A} < \frac{(\beta_A - \beta_B)^2}{V_A}$$

$$\iff \frac{(1 - \omega_A)(1 + \omega_A)}{(1 - \omega_A)^2} - \frac{V_B}{V_A} < \frac{(\beta_A - \beta_B)^2}{V_A}$$

$$\iff \frac{1 + \omega_A}{1 - \omega_A} - \frac{V_B}{V_A} < \frac{(\beta_A - \beta_B)^2}{V_A}$$

as is stated in *Equation 6*.

## Simulation of GxDiet effects on longevity in *Drosophila*

In *Figure 4*, we compare the effect estimates of Pallares et al. to ones we got in simulations of pervasive amplification. Here, we detail the simulation approach. We first generated true effects under each diet. For variants $j = 1, \ldots, 50,000$, we sampled a true effect under the high-sugar diet ($\beta_{h_j}$) and under the control diet ($\beta_{c_j}$). A random 60% of variants were set to have no effect under either diet, with the effects of the remaining 40% of variants sampled as

$$\begin{bmatrix} \beta_{c_j} \\ \beta_{h_j} \end{bmatrix} \sim \mathcal{N}\left( \begin{bmatrix} -0.125 \\ -0.15 \end{bmatrix}, 0.01 \cdot \begin{bmatrix} 1 & 1.4 \\ 1.4 & 1.96 \end{bmatrix} \right).$$

This corresponds to a systematic amplification of $1.4\times$ in the high-sugar compared to the control diet. We selected these parameters based on inspection of the resulting distribution of effects and their correspondence to the Pallares et al. data. We then simulated the effect estimation. For each variant, the effect estimate was simulated as normally distributed with mean equal to the true effect and standard deviation equal to a randomly sampled (with replacement) standard error from the effect estimates of Pallares et al. That is, given the simulated values of the true effect estimates $\beta_{c_j}$ and $\beta_{h_j}$, we simulated effect estimates as

$$\begin{bmatrix} \hat{\beta}_{c_j} \\ \hat{\beta}_{h_j} \end{bmatrix} \sim \mathcal{N}\left( \begin{bmatrix} \beta_{c_j} \\ \beta_{h_j} \end{bmatrix}, \begin{bmatrix} \hat{s}_{c_k}^2 & 0 \\ 0 & \hat{s}_{h_k}^2 \end{bmatrix} \right),$$

where $k$ represents the index of a randomly selected variant from the empirical data of Pallares et al. and $\hat{s}_{c_k}$ and $\hat{s}_{h_k}$ are the corresponding estimated standard errors for the effect estimates in the control and high-sugar groups, respectively. This process yielded vectors of estimated effects in the high-sugar group and control group, $\hat{\boldsymbol{\beta}}_h$ and $\hat{\boldsymbol{\beta}}_c$, respectively, and vectors of estimated standard errors in the high-sugar group and control group, $\hat{\boldsymbol{s}}_h$ and $\hat{\boldsymbol{s}}_c$, respectively. We then performed a Z-test for each variant under each diet, yielding two vectors of p-values, $\boldsymbol{p}_h$ and $\boldsymbol{p}_c$, corresponding to the high-sugar and control diets, respectively. Using these p-values, we followed a similar approach to Pallares et al. to classify the variants (*Figure 4D*). First, as in Pallares et al., we computed q-values separately for each diet (*Storey, 2003*), yielding $\boldsymbol{q}_h$ and $\boldsymbol{q}_c$, corresponding to the q-values of non-zero effects in the high-sugar and control diets, respectively. Then, we employed the following classification scheme for each variant $j = 1, \ldots, 50,000$:

1. If $q_{h_j} \geq 0.01$ and $q_{c_j} \geq 0.01 \rightarrow$ classify as *no effect*.
2. If $q_{h_j} < 0.01$ and $p_{c_j} \geq 0.1 \rightarrow$ classify as *high-sugar-specific effect*.
3. If $q_{c_j} < 0.01$ and $p_{h_j} \geq 0.1 \rightarrow$ classify as *control-specific effect*.
4. If $q_{c_j} < 0.01$ and $q_{h_j} < 0.01 \rightarrow$ classify as *shared effect*.

We note that p-value and q-value cutoffs used are nominally different than those used in the Pallares et al. study.

## Polygenic score simulations

In **Figure 5**, we show the results of multiple simulations where we compute polygenic scores in each of two contexts under amplification. Here, we detail the generation of data in the simulations and the methods for constructing polygenic scores. As in Results and discussion, we assumed that we have $n + m$ observations of a continuous trait, where the first $n$ individuals are observed in context $A$ and the final $m$ are observed in context $B$. For convenience, in this case we assumed $n = m$. Now, for variants $j = 1, \ldots, p$ we generated true effects in contexts $A$ and $B$ independently from the mixture model

$$\begin{bmatrix} \beta_{A_j} \\ \beta_{B_j} \end{bmatrix} \sim \pi_0 \delta_0 + (1 - \pi_0)\left( \alpha \mathcal{N}\left( \begin{bmatrix} 0 \\ 0 \end{bmatrix}, \begin{bmatrix} 1 & 1 \\ 1 & 1 \end{bmatrix} \right) \right) + (1 - \alpha)\mathcal{N}\left( \begin{bmatrix} 0 \\ 0 \end{bmatrix}, \begin{bmatrix} \frac{3}{2} & 1 \\ 1 & \frac{2}{3} \end{bmatrix} \right) \right),$$

where $\pi_0$ (which we set to 0.5) represents the proportion of SNPs with null effects in both contexts, $\alpha$ represents the proportion of non-null SNPs which have exactly equal effects in both contexts, and $1 - \alpha$ is the proportion of non-null SNPs which are generated as perfectly correlated but with $1.5\times$ the standard deviation in context A. Let $\vec{\beta}_A$ and $\vec{\beta}_B$ represent the resulting $p$-vectors of true effects for contexts $A$ and $B$, respectively. Next, we generated genotype counts for each of the $n + m$ individuals at all $p$ variants. Specifically, we independently generated genotypes as

$$f_j \sim \frac{1}{2} Beta(s_1, s_2) \text{ for } j = 1, \ldots, p$$
$$g_{ij} \sim Binomial(2, f_j) \text{ for } i = 1, \ldots, n + m,$$

where $f_j$ is the minor allele frequency at variant $j$ in the population, $s_1$ and $s_2$ are parameters controlling the distribution of minor allele frequencies in the population, and $g_{ij}$ is the observed genotype for individual $i$ at variant $j$. Here, we set $s_1 = 1$ and $s_2 = 5$. Let $G_A$ and $G_B$ represent the generated $n \times p$ matrices of genotypes in contexts $A$ and $B$, respectively. Finally, we generated the observed continuous traits for context $A$ ($\vec{y}_A$) and context $B$ ($\vec{y}_B$) as $\vec{y}_A \vec{y}_B$

$$\vec{y}A \sim \mathcal{N}\left( G_A \vec{\beta}_A, \sigma_A^2 I_n \right)$$
$$\vec{y}B \sim \mathcal{N}\left( G_B \vec{\beta}_B, \sigma_B^2 I_m \right),$$

where $\sigma_A^2$ and $\sigma_B^2$ are the observation variances in contexts $A$ and $B$, respectively, and $I_w$ is the $w \times w$ identity matrix. In our simulations, we set $\sigma_A^2$ and $\sigma_B^2$ such that the narrow sense heritability is 40% in each context. So that we may later test the accuracy of our polygenic scores, we generated both a training set (consisting of $n$ individuals in each context, where $n=1000$ in the low power simulation and $n = 50,000$ in the high power simulation) for effect estimation and a test set (consisting of 3000 individuals in each context) using the above distributions. **Figure 5** compares four distinct approaches for constructing polygenic scores, derived from three allelic effect estimation approaches: additive estimation with shrinkage, GxE estimation with shrinkage, and *mash*. First, the additive and GxE estimates are derived independently for each variant as described in Results and discussion. Let $\hat{\beta}_A$ and $\hat{\beta}_B$ be the $p$-vectors of GxE estimates of effects in context $A$ and $B$, respectively. Similarly, let $\hat{s}_A$ and $\hat{s}_B$ be the $p$-vectors of the standard errors of GxE estimates of effects in context $A$ and $B$, respectively. Finally, let $\hat{\beta}_{A \cup B}$ be the $p$-vector of estimated effects from the additive model and $\hat{s}_{A \cup B}$ be the $p$-vector of standard errors of estimated effects from the additive model. Using the GxE estimates, we also constructed estimates of the effects in each context using *mash*. Specifically, we ran mash on the $n \times 2$ matrices $\begin{bmatrix} \hat{\beta}_A & \hat{\beta}_B \end{bmatrix}$ (of effects) and $\begin{bmatrix} \hat{s}_A & \hat{s}_B \end{bmatrix}$ (of standard errors). *mash* then yields $p(\vec{\beta}_A | \hat{\beta}_A, \hat{s}_A)$ and $p(\vec{\beta}_B | \hat{\beta}_B, \hat{s}_B)$, the posterior distributions of the effects in contexts $A$ and $B$, respectively. To construct each polygenic score, we made two choices. First, a choice between the three sets of p-values (or pseudo p-values, see below) for thresholding—we include the 833 (corresponding to one-third of the causal variants) most significant variants in the polygenic

score. The second choice was between the three sets of effect estimates to be used as weights in the polygenic score (*Figure 5*). For instance, when the GxE model was used for ascertainment, we selected the set of variants $\Omega_A \subset \{1, \ldots, p\}$ consisting of the variants with the 833 smallest p-values and $\Omega_B \subset \{1, \ldots, p\}$ consisting of the variants with the 833 smallest p-values (derived from $\hat{\beta}_B$ and $\hat{s}_B$). Then, we predicted trait values (out of sample) by multiplying the effect estimates of our chosen 'estimation method' (for *mash* we use the posterior mean) by the effect allele count at each of the selected variants for the individual in question.

## Acknowledgements

We thank Doc Edge, Marc Feldman, Mark Kirkpatrick, Molly Przeworski, Anil Raj, Elliot Tucker-Drob, and members of the Harpak Lab for comments on the manuscript. We thank Peter Andolfatto, Julien Ayroles, and Tom Juenger for helpful discussions. All authors were supported by NIH R35GM151108 to AH. SP Smith was also supported by NIH RF1AG073593. This study was conducted using the UK Biobank resource under application 61666, as approved by the University of Texas at Austin institutional review board (protocol 2019-02-0125).

## Additional information

### Funding

| Funder | Grant reference number | Author |
|---|---|---|
| National Institutes of Health | R35GM151108 | Arbel Harpak |
| National Institutes of Health | RF1AG073593 | Samuel Pattillo Smith |
| Pew Charitable Trusts | Pew Biomedical Scholarship | Arbel Harpak |

The funders had no role in study design, data collection and interpretation, or the decision to submit the work for publication.

### Author contributions

Eric Weine, Formal analysis, Investigation, Visualization, Methodology, Writing – original draft, Writing – review and editing; Samuel Pattillo Smith, Formal analysis, Investigation, Methodology, Writing – review and editing; Rebecca Kathryn Knowlton, Data curation, Formal analysis, Investigation, Methodology; Arbel Harpak, Conceptualization, Formal analysis, Supervision, Funding acquisition, Investigation, Visualization, Methodology, Writing – original draft, Project administration, Writing – review and editing

### Author ORCIDs

Eric Weine ⓘ https://orcid.org/0009-0001-7809-1649
Samuel Pattillo Smith ⓘ https://orcid.org/0000-0002-6269-0276
Arbel Harpak ⓘ https://orcid.org/0000-0002-3655-748X

Reviewer #1 (Public review): https://doi.org/10.7554/eLife.99210.3.sa1
Author response https://doi.org/10.7554/eLife.99210.3.sa2

## Additional files

### Supplementary files
MDAR checklist

### Data availability
All data used in this work was available via previously published studies.

The following previously published datasets were used:

| Author(s) | Year | Dataset title | Dataset URL | Database and Identifier |
|---|---|---|---|---|
| Zhu C, Ming MJ, Cole JM, Edge MD, Kirkpatrick M, Harpak A | 2023 | Additive summary statistics | https://doi.org/10.5281/zenodo.7508246 | Zenodo, 10.5281/zenodo.7508246 |
| Pallares LF, Lea AJ, Han C, Filippova EV, Andolfatto P, Ayroles JF | 2022 | Dietary stress remodels the genetic architecture of lifespan variation in outbred *Drosophila* | https://www.ncbi.nlm.nih.gov/bioproject/?term=PRJNA725602 | NCBI BioProject, PRJNA725602 |
| Zhu C, Ming MJ, Cole JM, Edge MD, Kirkpatrick M, Harpak A | 2022 | Sex-specific summary statistics | https://doi.org/10.5281/zenodo.7222725 | Zenodo, 10.5281/zenodo.7222725 |

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

## Appendix 1

### 1 Validating and applying the decision rule to real data

To validate the decision rule of *Equations 4–6* of the main text, we considered the example of sex as a context for physiological traits using UK Biobank data (*Bycroft et al., 2018*). For each of 27 physiological quantitative traits, we first split the sample by sex chromosome karyotypes into XX individuals (females) and XY individuals (males). For each sex, we randomly split the sample into a training set with 80% of the individuals and a test set with 20% of the individuals. We estimated within-sex GWAS in the training and test sets separately. We denote by $\hat{\beta}_{Z_i}^{(t)}$ the marginal effect estimate of the $i$th SNP in sex $Z$ in set $t$, and by $\hat{s}_{Z_i}^{(t)}$ the corresponding standard error. We denote by $\hat{\beta}_{M \cup F_i}^{(train)}$ the estimated effect of the $i$th SNP in the union of training sets (with both sexes included). In our validation procedure, we treated $\hat{\beta}_{M_i}^{(test)}$ as the 'ground-truth', i.e., as the true effect. While $\hat{\beta}_{M_i}^{(test)}$ is of course a very noisy estimate of $\beta_{M_i}$, it is an unbiased estimate, so we expect that estimates that are closer to $\hat{\beta}_{M_i}^{(test)}$ in the aggregate will also be closer to $\beta_{M_i}$. Using this assumption, we define two empirical quantities,

$$error(\hat{\beta}_{M_i}^{(train)}) = (\hat{\beta}_{M_i}^{(train)} - \hat{\beta}_{M_i}^{(test)})^2$$

$$error(\hat{\beta}_{M \cup F_i}^{(train)}) = (\hat{\beta}_{M \cup F_i}^{(train)} - \hat{\beta}_{M_i}^{(test)})^2.$$

Given $\sqrt{V_{M_i}}$ and $|\beta_{M_i} - \beta_{F_i}|$, *Equation 4* allows us to calculate the expected difference between $error(\hat{\beta}_{M_i}^{(train)})$ and $error(\hat{\beta}_{M \cup F_i}^{(train)})$. However, we do not observe $\sqrt{V_{M_i}}$ or $|\beta_{M_i} - \beta_{F_i}|$. Instead, we estimated these quantities and examined the agreement of the estimates in the two sets. We estimated $\sqrt{V_{M_i}}$ as $\hat{s}_{M_i}^{(train)}$. Estimating $|\beta_{M_i} - \beta_{F_i}|$ is more difficult, as the intuitive $|\hat{\beta}_{M_i}^{(train)} - \hat{\beta}_{F_i}^{(train)}|$ is an upwardly biased estimator. Instead, we used an empirical shrinkage estimator (*ash Stephens, 2017*) to estimate $|\beta_{M_i} - \beta_{F_i}|$. In essence, *ash* takes in a vector of effect estimates $\hat{\beta}$ and a vector of corresponding standard errors $\hat{s}$, estimates a prior distribution of the true effects , and then uses this prior to obtain a posterior distribution of true effects. Specifically, for a large random sample of sites, $i = 1, \ldots, p$, we ran *ash* on the vectors

$$\hat{\boldsymbol{\beta}} = \left(\hat{\beta}_{M_1}^{(train)} - \hat{\beta}_{F_1}^{(train)}, \ldots, \hat{\beta}_{M_p}^{(train)} - \hat{\beta}_{F_p}^{(train)}\right)$$

$$\hat{s} = \left(\sqrt{\left(\hat{s}_{M_1}^{(train)}\right)^2 + \left(\hat{s}_{F_1}^{(train)}\right)^2}, ..., \sqrt{\left(\hat{s}_{M_p}^{(train)}\right)^2 + \left(\hat{s}_{F_p}^{(train)}\right)^2}\right).$$

and took the absolute value of *ash*'s output posterior mean estimates of $\beta_{M_i} - \beta_{F_i}$ as our estimate of $|\beta_{M_i} - \beta_{F_i}|$ for each site. With our estimates of $|\beta_{M_i} - \beta_{F_i}|$ and $\sqrt{V_{M_i}}$ in hand, we calculated the expected difference in squared errors using *Equation 4* (x-axis in *Figure 3A*) and compared it with the actual difference between $error(\hat{\beta}_{M_i}^{(train)})$ and $error(\hat{\beta}_{M \cup F_i}^{(train)})$ (*Figure 1*). For *Figure 3A and B*, we used the same procedure as above to estimate the x and y axes, except we did not employ any data splitting.

### 2 Equivalence of regression coefficient estimates for explicit interaction term and stratified model

In the main text, we discuss the estimation of context-dependent effects through the estimation of two effects in subsamples stratified by context. The generative model defined in *Equation 1* is

$$y_i \sim \begin{cases} \mathcal{N}(\alpha_A + \beta_A g_i, \sigma_A^2) & \text{if } i \in \{1, \ldots, n\} \\ \mathcal{N}(\alpha_B + \beta_B g_i, \sigma_B^2) & \text{if } i \in \{n+1, \ldots, n+m\}, \end{cases} \tag{S1}$$

where $\beta_A$ and $\beta_B$ are fixed, context-specific effects of a reference allele at a biallelic, autosomal variant $i$, $g_i \in \{0, 1, 2\}$ is the observed reference allele count. $\alpha_A$ and $\alpha_B$ are the context-specific intercepts, corresponding to the mean trait value for individuals with zero reference alleles in context $A$ and $B$, respectively. $\sigma_A^2$ and $\sigma_B^2$ are context-specific observation variances. Another common way to define a generative GxE model is with a linear genotype-by-context interaction term, namely,

$$y_i \sim \mathcal{N}(\alpha + \beta_G g_i + \beta_E e_i + \beta_{GxE} g_i e_i, \sigma^2) \text{ independently for } i = 1, \ldots, n + m, \tag{S2}$$

where $\alpha$ is the mean trait value for individuals with zero reference alleles in context $A$, $\beta_G$ is the genetic effect, $\beta_E$ is the contextual effect, $\beta_{GxE}$ in the genotype-context interaction term, $e_i$ is the contextual covariate for individual $i$, defined to be 0 if the individual is in context $A$ and 1 if the individual is in context $B$, and $\sigma^2$ is the homoskedastic noise term. Here, we show that least squares estimation of the two models above context-specific effect estimates. The OLS estimate of the intercept and context-specific allelic effects under the model described in *Equation S1* are

$$\hat{\alpha}_A, \hat{\beta}_A = \mathrm{argmin}_{\alpha_A, \beta_A} \sum_{i=1}^{n} \left(y_i - (\alpha_A + \beta_A g_i)\right)^2 \tag{S3}$$

$$\hat{\alpha}_B, \hat{\beta}_B = \mathrm{argmin}_{\alpha_B, \beta_B} \sum_{i=n+1}^{n+m} \left(y_i - (\alpha_B + \beta_B g_i)\right)^2. \tag{S4}$$

The OLS estimates under the model described in *Equation S2* are

$$\hat{\alpha}, \hat{\beta}_G, \hat{\beta}_E, \hat{\beta}_{GxE} = \mathrm{argmin}_{\alpha, \beta_G, \beta_E, \beta_{GxE}} \sum_{i=1}^{n+m} \left(y_i - (\alpha + \beta_G g_i + \beta_E e_i + \beta_{GxE} g_i e_i)\right)^2. \tag{S5}$$

Since $e_i = 1$ if individual $i$ is in context $B$ and 0 otherwise, we can re-write *Equation S5* as

$$\hat{\alpha}, \hat{\beta}_G, \hat{\beta}_E, \hat{\beta}_{GxE} = \mathrm{argmin}_{\alpha, \beta_G, \beta_E, \beta_{GxE}} \left( \sum_{i=1}^{n} \left(y_i - (\alpha + \beta_G g_i)\right)^2 + \sum_{i=n+1}^{n+m} \left(y_i - (\alpha + \beta_G g_i + \beta_E + \beta_{GxE} g_i)\right)^2 \right)$$

$$= \mathrm{argmin}_{\alpha, \beta_G, \beta_E, \beta_{GxE}} \left( \sum_{i=1}^{n} \left(y_i - (\alpha + \beta_G g_i)\right)^2 + \sum_{i=n+1}^{n+m} \left(y_i - ((\alpha + \beta_E) + (\beta_G + \beta_{GxE}) g_i)\right)^2 \right).$$

By *Equation S3* we have that $\sum_{i=1}^{n} \left(y_i - (\alpha + \beta_G g_i)\right)^2$ is minimized by setting $\alpha = \hat{\alpha}_A$ and $\beta_G = \hat{\beta}_A$. In turn, by *Equation S4* we have that $\sum_{i=n+1}^{n+m} \left(y_i - ((\alpha + \beta_E) + (\beta_G + \beta_{GxE}) g_i)\right)^2$ is minimized by setting $\alpha + \beta_E = \hat{\alpha}_B$ and setting $\beta_G + \beta_{GxE} = \hat{\beta}_B$. Thus, both terms are simultaneously minimized by setting

$$\alpha = \hat{\alpha}_A$$

$$\beta_G = \hat{\beta}_A$$

$$\beta_E = \hat{\alpha}_B - \hat{\alpha}_A$$

$$\beta_{GxE} = \hat{\beta}_B - \hat{\beta}_A.$$

Then, under the explicit interaction model, for an individual in context $A$ the estimated intercept is $\hat{\alpha}_A$ and the estimated genetic effect is $\hat{\beta}_A$. And, for an individual in context $B$, the estimated intercept is $\hat{\alpha}_A + \hat{\alpha}_B - \hat{\alpha}_A = \hat{\alpha}_B$ and the estimated genetic effect is $\hat{\beta}_A + \hat{\beta}_B - \hat{\beta}_A = \hat{\beta}_B$. Thus, conditional on the context, both the explicit interaction model and the stratified model provide the same effect and intercept estimates. We note that there is still an important difference between the stratified and explicit interaction models, as the stratified model allows for heteroskedasticity between contexts where the explicit interaction model does not. We also note that the proof of equivalence relies on the context variable being binary.

## 3 Continuous context variable

Here, we extend our analysis of single-site estimation in a binary context to the case of a continuous context variable. Specifically, for individuals indexed by $i = 1, \ldots, n$, we consider the generative model

$$y_i \sim \mathcal{N}(\beta_0 + \beta_G g_i + \beta_E e_i + \beta_{G \times E} g_i e_i, \sigma^2), \tag{S6}$$

where $y_i \in \mathbb{R}$ is the continuous trait value for individual $i$, $g_i \in \{0, 1, 2\}$ is the number of reference alleles carried by a diploid individual $i$ at a focal biallelic variant of interest, $e_i \in \mathbb{R}$ is the continuous context for individual $i$, $\beta_0$ is the the mean trait value for individuals with the context variable value of zero and zero reference alleles, $\beta_G$ is the main (additive) allelic effect, $\beta_E$ is the effect of the context, $\beta_{G \times E}$ is the main interaction effect, and $\sigma^2$ is the residual variance. For simplicity, we will assume $\beta_0 = 0$. For a binary context, we considered the difference in MSE of the 'additive model' and 'GxE model' in estimating the context-specific genetic effect. Analogously, for a continuous context we will consider the difference in MSE in estimating the total genetic effect conditional on a particular value of the context variable. To perform this analysis, we will consider the estimation of two models. The first model, which we will call the 'GxE' model, is described in **Equation S6** above (where we omit the intercept because we assume it is 0). For convenience, we will use the following matrix notation:

$$y = \begin{bmatrix} y_1 \\ \vdots \\ y_n \end{bmatrix}, X = \begin{bmatrix} g_1 & e_1 & g_1 e_1 \\ \vdots & \vdots & \vdots \\ g_n & e_n & g_n e_n \end{bmatrix}, \beta = \begin{bmatrix} \beta_G \\ \beta_E \\ \beta_{G \times E} \end{bmatrix}.$$

Then, we consider the standard least squares estimate of $\hat{\beta} = (X^T X)^{-1} X^T y$, for which

$$\hat{\beta} \sim \mathcal{N}(\beta, \sigma^2 (X^T X)^{-1}). \tag{S7}$$

We will assume that the vector of genotypes $\vec{g}$ and the vector of context covariates $\vec{e}$ are orthogonal in our sample and are both mean centered. (In the following section, we discuss the case of gene-context correlation, where these assumptions are not met.) Then, the matrix $X^T X$ will be approximately diagonal. Under this approximation, we can write **Equation S7** as

$$\hat{\beta} \sim \mathcal{N}(\beta, \Sigma),$$

where

$$\Sigma \approx \sigma^2 \text{diag}\left( \frac{1}{\sum_{i=1}^{n} g_i^2}, \frac{1}{\sum_{i=1}^{n} e_i^2}, \frac{1}{\sum_{i=1}^{n} g_i^2 e_i^2} \right). \tag{S8}$$

Now, we are interested in the error of the GxE model in estimating the genetic effect conditional on a particular value of the context $e$. By **Equation S6**, we define the true genetic effect conditional on a particular value of the context $e$ as $\beta_G + \beta_{G \times E} e$. Thus, to evaluate the error of the GxE model, the conditional expected error is

$$E[((\hat{\beta}_G + \hat{\beta}_{G \times E} e) - (\beta_G + \beta_{G \times E} e))^2 | e]. \tag{S9}$$

By the definition of conditional variance, we can write **Equation S9** as

$$Var[(\beta_G + \beta_{G \times E} e) - (\hat{\beta}_G + \hat{\beta}_{G \times E} e) | e] + (E[(\beta_G + \beta_{G \times E} e) - (\hat{\beta}_G + \hat{\beta}_{G \times E} e) | e])^2$$

$$= Var[(\beta_G + \beta_{G \times E} e) - (\hat{\beta}_G + \hat{\beta}_{G \times E} e) | e] \quad \text{(since } \hat{\beta}_G \text{ and } \hat{\beta}_{G \times E} \text{ are unbiased estimates)}$$

$$= Var[\hat{\beta}_G + \hat{\beta}_{G \times E} e | e] \quad \text{(since } \beta_G \text{ and } \beta_{G \times E} \text{ are fixed parameters)}$$

$$\approx \frac{\sigma^2}{\sum_{i=1}^{n} g_i^2} + \frac{\sigma^2}{\sum_{i=1}^{n} g_i^2 e_i^2} e^2$$

(by **Equation S8**).

We will also consider the estimation error associated with the 'additive model', which in this case estimates the model described in **Equation S6** assuming that $\beta_{G \times E} = 0$. In effect, this assumes that the genetic effect is equal across all values of the context. Because $X^T X$ is approximately diagonal matrix, removing the $G \times E$ covariate and regression coefficient has a negligible effect on both the least squares estimates of $\beta_G$ and its sampling distribution. Thus, we can approximate the error of the additive model by calculating the conditional expectation

$$E[((\beta_G + \beta_{G \times E} e) - \hat{\beta}_G)^2 | e] \tag{S10}$$

under the model in *Equation S6*. Again, by the definition of the conditional expectation, we can write *Equation S10* as

$$Var[(\beta_G + \beta_{G \times E} e) - \hat{\beta}_G | e] + (E[(\beta_G + \beta_{G \times E} e) - \hat{\beta}_G | e])^2$$
$$= \frac{\sigma^2}{\sum_{i=1}^n g_i^2} + \beta_{G \times E}^2 e^2.$$

Thus, conditional on a particular value of $e$, we prefer the $G \times E$ estimator for estimating the genetic effect if and only if

$$\frac{\sigma^2}{\sum_{i=1}^n g_i^2} + \frac{\sigma^2}{\sum_{i=1}^n g_i^2 e_i^2} e^2 < \frac{\sigma^2}{\sum_{i=1}^n g_i^2} + \beta_{G \times E}^2 e^2 \tag{S11}$$

$$\iff \frac{\sigma^2}{\sum_{i=1}^n g_i^2 e_i^2} e^2 - \beta_{G \times E}^2 e^2 < 0 \tag{S12}$$

$$\iff \frac{\sigma^2}{\sum_{i=1}^n g_i^2 e_i^2} < \beta_{G \times E}^2. \tag{S13}$$

This decision rule has a very intuitive interpretation: the $G \times E$ model is preferable when the estimation variance of the interaction term is smaller than the squared interaction effect size. While the exact difference in MSE between the additive and $G \times E$ models depends on the value of the context variable (see *Equation S11*), the decision rule (*Equation S13*) does not.

## 4 Gene-context correlation

The simple decision rule given in *Equation S13* was derived using the assumption that the observed genotype vector and the observed context vector were orthogonal. In statistical terms, one may expect this condition to approximately hold when sampling from a population where $r(G, E)$, the correlation between genotypes and the context, is 0. In general, this assumption may not be reasonable. When this assumption is violated, the design matrix $X$ will no longer form a diagonal matrix when left-multiplied by its transpose. This results in both (a) larger standard errors for each of the marginal coefficients and (b) non-zero sampling covariance between the coefficients. In addition, $|r(G, E)| > 0$ may result in omitted variable bias in the additive model. As a result, the decision rule of *Equation S13* will not hold exactly under conditions of $|r(G, E)| > 0$, and in fact may be far from correct when $|r(G, E)|$ is large. In general, creating a closed-form decision rule to choose between the GxE and additive models in this scenario is not possible. However, in this section we use a simulation study to gain intuition about the change in bias-variance trade-off in the presence of gene-context correlation. In our simulations, for a group of $n$ individuals, we generate genotypes as

$$g_1, \ldots, g_n \stackrel{iid}{\sim} Binomial(2, p).$$

Then, for some constant $c$, we generate context covariates as

$$e_i \sim \mathcal{N}(c \cdot g_i, \sigma_E^2)$$

independently for $i = 1, \ldots, n$. Finally, we generate trait values $y_1, \ldots, y_n$ using *Equation S6* for particular values of the remaining parameters. To explore how the bias-variance trade-off changes for different values of $r(G, E)$, we ran a set of simulations with different values of $c$ in the range of $-3.5$ to $3.5$. For each of these simulations, we generated a dataset of $n = 250$ individuals with $\sigma_E^2 = 1$, $p = 0.1$, $\beta_0 = 0$, $\beta_G = 0.25$, $\beta_E = 0$, $\beta_{G \times E} = -.125$, and $\sigma_y^2 = 7$. The results of the simulation are shown in *Appendix 1—figure 6*. Non-zero $r(G, E)$ had a number of consequences. First, as the magnitude of $r(G, E)$ increases, the magnitude of the bias of the genetic effect estimate of the additive model also increases (*Appendix 1—figure 6A*). In our simulation, we set the genetic effect and the interaction effect to have opposite signs. When $r(G, E)$ is positive, it pulls the genetic effect estimate toward the interaction effect. In contrast, when $r(G, E)$ is negative, it pulls the genetic effect estimate away from the interaction effect. Since the $G \times E$ model includes all causal variables, it is unbiased. Second, as the magnitude of $r(G, E)$ increases, the variance of the genetic effect estimates of both the additive and $G \times E$ models increase (*Appendix 1—figure 6B*). This is

because when variables in a regression are (positively or negatively) correlated, it is more difficult to distinguish between their effects. The $G \times E$ model appears to be affected more strongly here, which likely results from the inclusion of an additional interaction term that is highly correlated with both the genotype covariate and the context covariate. Third, the interaction term in the $G \times E$ model remains unbiased under $r(G,E)$, and increases in variance with larger magnitude of $r(G,E)$ (*Appendix 1—figure 6C and D*).

## 5 Classification-based inference in Pallares et al., 2022

In the main text, we discuss the characterization of GxE by Pallares et al., based on significance testing under each of the diets. The authors classified variants according to whether or not their associations with survivorship were significant under each diet as follows:

1. Significant under neither diet → classify as *no effect*.
2. Significant when fed the high-sugar diet, but not when fed the control diet → classify as *high-sugar-specific effect*.
3. Significant when fed the control diet, but not when fed the high-sugar diet → classify as *control-specific effect*.
4. Significant under both diets → classify as *shared effect*.

Approximately 31% were high-sugar specific, while the remaining 69% of the variants were shared. Fewer than 1% were labeled as having control-specific effects. The authors concluded that high-sugar-specific effects on longevity are pervasive, compatible with the hypothesis of widespread cryptic genetic variation for longevity. This 'top hits' approach places an emphasis on the context(s) in which trait associations are statistically significant, rather than on estimating how the context-specific allelic effects covary. In addition, this particular classification system also does not cover all possible ways in which context-specific effects may differ. A key example is the case where true effects are concordant in sign but differ in magnitude. The strong positive covariance of estimated effects observed genome-wide (*Figure 4A*) suggests this case merits consideration. Such variants may fall into each of the existing categories, depending on the magnitude of effects and statistical power in each of the contexts. Three of the four possible classifications are clearly wrong, but what about the 'shared' category? The class 'shared' may be interpreted as suggesting lack of context dependency (Figure 1C in *Pallares et al., 2023*). However, it will tend to include variants having strong effects under both diets, regardless of whether or not the diet-specific effects are similar. As Pallares et al. also note, there are marked differences in the magnitude of diet-specific estimated effects of variants in the 'shared' category. Among the approximately 1500 variants labeled as shared, the estimated effect under the high-sugar diet is on average about $1.3\times$ that of the estimated effect in the control diet. Notably, the classification as diet-specific does not imply that a variant has an unusually large effect under this diet. On average, variants classified as shared actually have a slightly larger estimated effect under the high-sugar diet than variants classified as high-sugar specific (t-test p=$1.6 \times 10^{-5}$). Thus, instead of suggesting little-to-no effect under the control diet and a large effect under the high-sugar diet (as predicted by the hypothesis that cryptic genetic variation is pervasive), the classification as high-sugar specific may commonly just point us to intermediate size effects—large enough to be significant in the systematically larger effects context (where power is higher) yet too small to be significant in the systematically smaller effects context (where power is lower).

## 6 Re-analysis of GxE in the Pallares et al. experiment

In the main text, we show that the classification of significantly associated variants in Pallares et al. is consistent with pervasive amplification, despite the fact that this mode of covariance was not one of the generative modes considered by the authors. Here, we re-estimate the modes of covariance of genetic effects under the two diets, using an approach adapted from the one we have previously used (*Zhu et al., 2023*). Specifically, we used the framework of 'multivariate adaptive shrinkage' (*mash*) to model the covariance of effects between the high-sugar and control diets across all variants. We fit a multivariate normal mixture model to the summary statistics of Pallares et al. (*Urbut et al., 2019*). In short, *mash* takes in a set of multidimensional effect estimates and standard errors and estimates the true underlying distribution of effects as a mixture of zero-centered multivariate normal distributions and a point mass at 0. Each mixture component in the model must be pre-specified before the fitting procedure. Following (*Zhu et al., 2023*), we pre-specified a dense grid of covariance matrices to be input to mash. In particular, each covariance matrix is of the form

$$\begin{bmatrix} \sigma_c^2 & \rho\sigma_{hs}^2\sigma_c^2 \\ \rho\sigma_{hs}^2\sigma_c^2 & \sigma_{hs}^2 \end{bmatrix},$$

where $\sigma_c^2$ is the variance of the true effects in the control group, $\sigma_{hs}^2$ is the variance of the true effects in the high-sugar group, and $\rho$ is the correlation between the effects in the high-sugar and control groups. To specify the set of covariance matrices, we formed a grid encompassing possible values of the correlation of allelic effects across diets and ratio of variances under each of the diets. Specifically, we varied $\rho$ in the set of values $(-1, -\frac{3}{4}, -\frac{1}{2}, -\frac{1}{4}, 0, \frac{1}{4}, \frac{1}{2}, \frac{3}{4}, 1)$, and we varied the ratio of the variances, $\frac{\sigma_c^2}{\sigma_{hs}^2}$ on a logarithmic scale in the range 0.5 and 1.5. Taking the Cartesian product of these sets yielded a grid of covariance matrices. Because the number of pre-specified covariance matrices was large, and to avoid overfitting, we used a forward stepwise selection procedure (**Hastie et al., 2009**). We first started with a model with only the null covariance matrix (i.e. no effect under either diet), and then added one covariance matrix to the mixture in a greedy manner in each step, by searching over the space of all covariance matrices for a matrix that maximally improves the likelihood of the mixture model. In each step, we either decide to include the new covariance matrix in the model and move to the next step, or instead stop the procedure and not include the new matrix, if the improvement in likelihood compared to the previous step was below a pre-specified threshold. This threshold was determined by conducting a level $\alpha = 0.05$ likelihood-ratio test. To mitigate possible linkage disequilibrium among the input SNPs, we performed the procedure described above on a subset roughly 12K out of the roughly 270K variants in the data. These SNPs each came as a random sample from 12K roughly evenly spaced chromosomal blocks (in terms of physical distance). Based on the output of this procedure—the mixture weights in the model ultimately chosen—92% of variants have non-zero effects under both diets but larger effects under the high-sugar diet (**Figure 4**). This suggests that instead of affecting just a small subset of variants, a high-sugar diet amplifies the effects of the vast majority of variants on lifespan. Moreover, *mash* assigned zero weight to covariance matrices where an effect is non-zero in one context but zero in another.

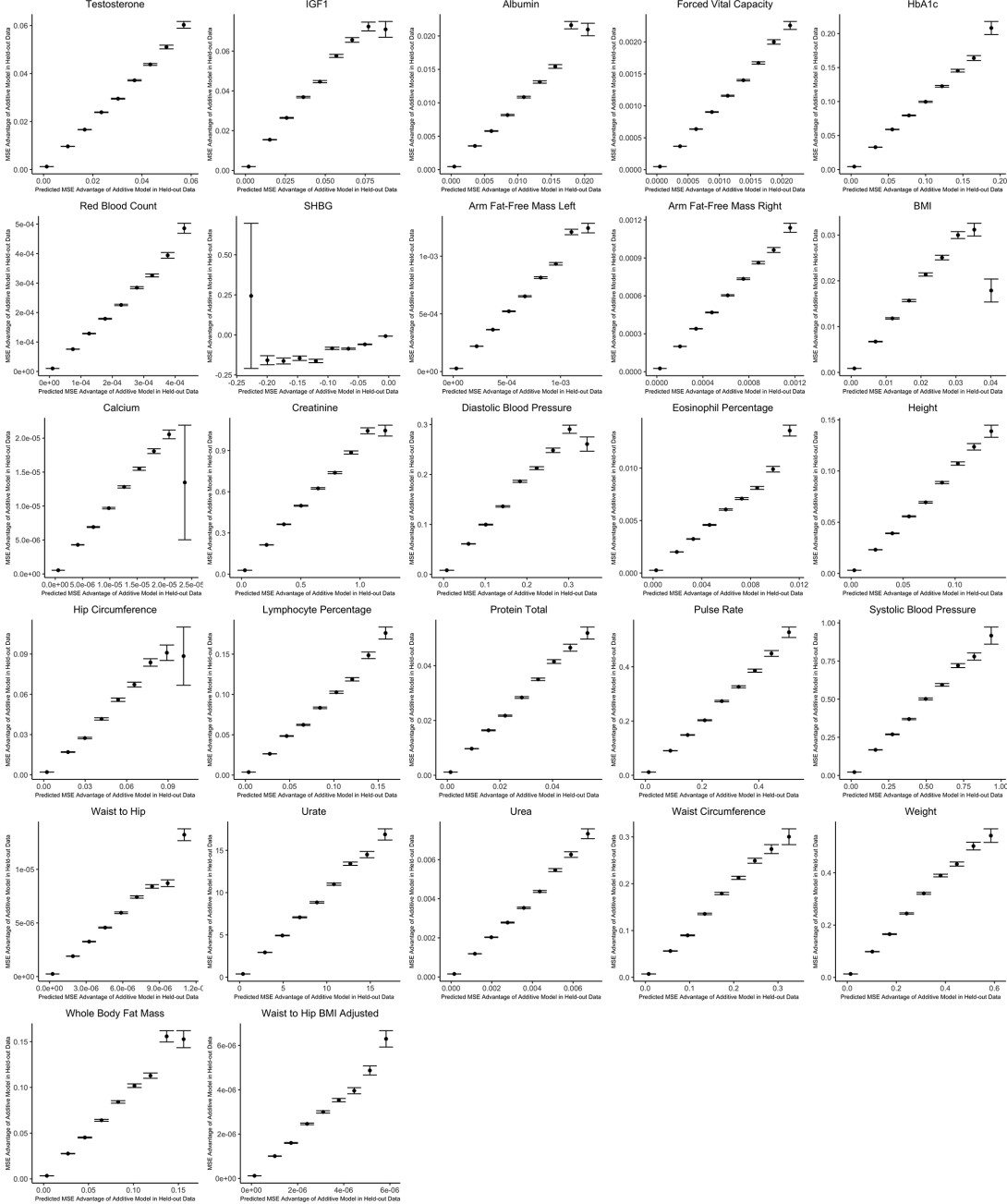

**Appendix 1—figure 1.** Out-of-sample validation of decision rule for sex-stratified genome-wide association studies (GWAS) for 27 complex traits. For each trait, we use the procedure specified in the section Validating and applying the decision rule to real data. The *x*-axis shows the expected mean squared error (MSE) advantage of the additive estimator in predicting the male effect based on the decision rule, where the *y*-axis shows the actual MSE advantage of the additive model in predicting the male effect calculated using an independent sample. Values on the *x*-axis are grouped into nine evenly spaced bins, and the *y*-axis shows bin averages (with error bars indicating 1 standard deviation, where the sample size is equal to the number of SNPs falling into each bin). For ease of viewing, data above the 99th percentile and below the 1st percentile on the *x*-axis are removed.

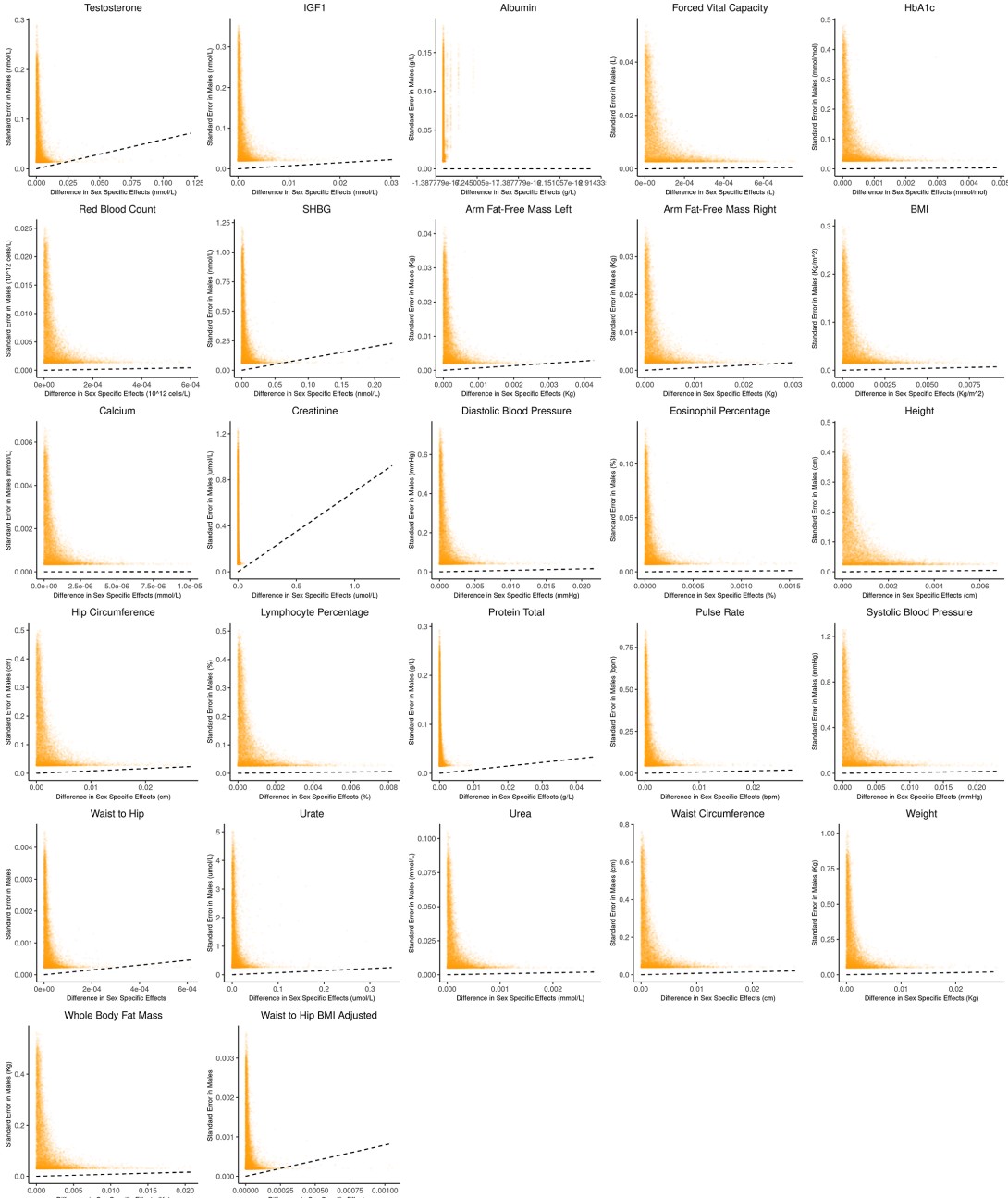

**Appendix 1—figure 2.** Bias-variance trade-off at random variants in a sex-stratified genome-wide association studies (GWAS). This figure extends *Figure 3A* of the main text to all physiological traits analyzed. The *x*-axis shows the estimated absolute difference between the effect of variants in males and females. The *y*-axis shows the measured standard error for each variant in males, the focal context here. The dashed line shows the decision boundary for effect estimation in males. The difference in mean squared error (MSE) between estimation methods increases linearly with distance from the dashed line, as in *Figure 2*. If a variant falls above (below) the line, the additive (gene-by-environment interaction [GxE]) estimator has a lower MSE.

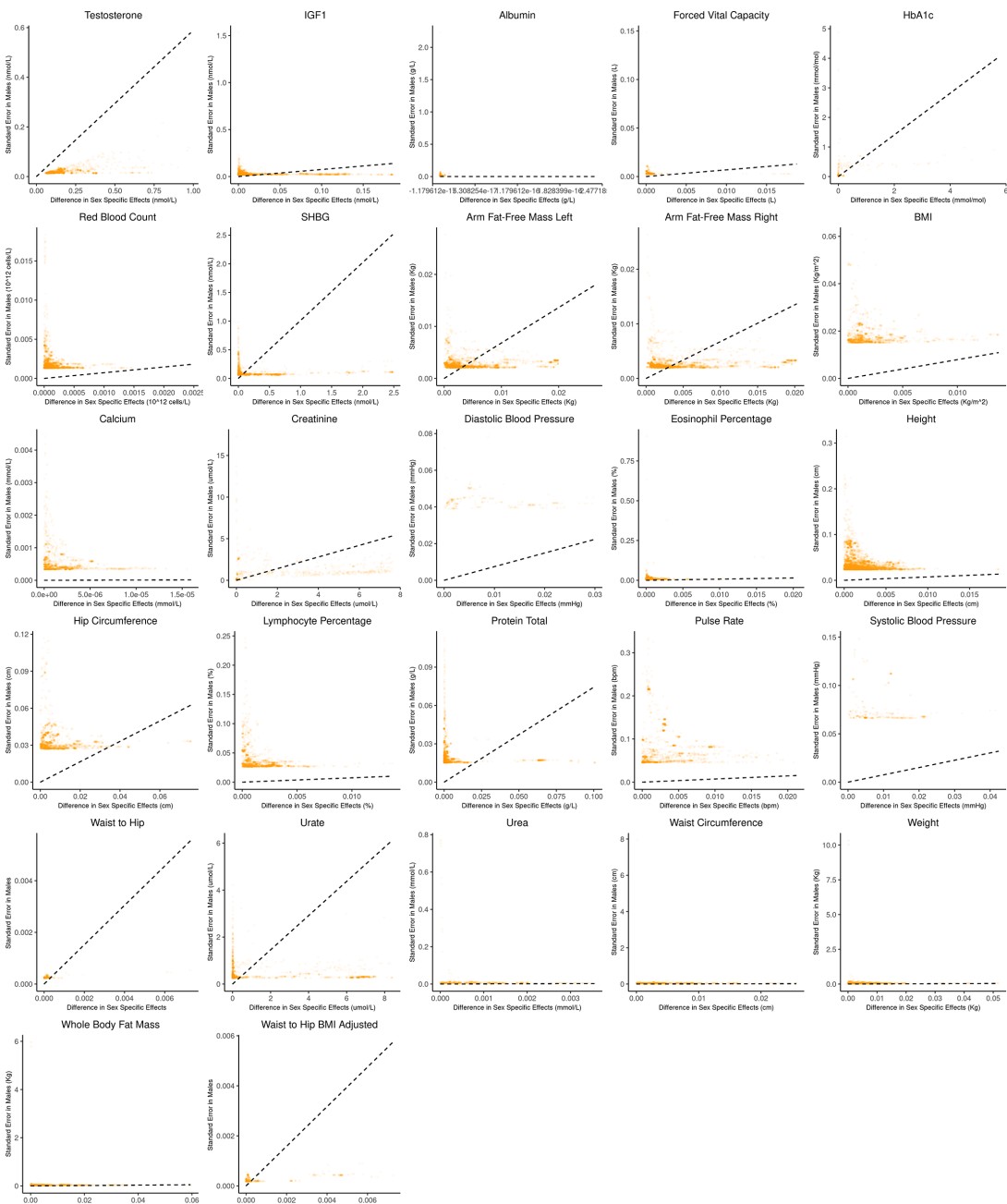

**Appendix 1—figure 3.** Bias-variance trade-off at genome-wide significant variants (p-value $< 5 \times 10^{-8}$ in males) in a sex-stratified genome-wide association studies (GWAS). This figure extends *Figure 3B* of the main text to all physiological traits analyzed. The x-axis shows the estimated absolute difference between the effect of variants in males and females. The y-axis shows the measured standard error for each variant in males, the focal context here. The dashed line shows the decision boundary for effect estimation in females. The difference in mean squared error (MSE) between estimation methods increases linearly with distance from the dashed line, as in *Figure 2*. If a variant falls above (below) the line, the additive (gene-by-environment interaction [GxE]) estimator has a lower MSE.

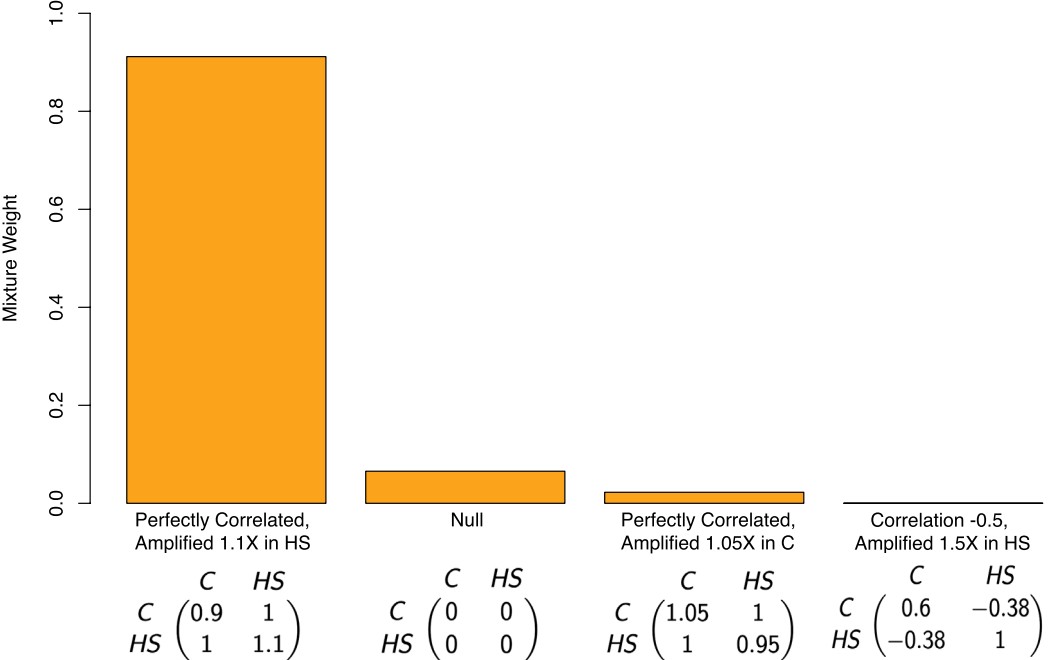

**Appendix 1—figure 4.** Estimated mixture weights on covariance of true effects in Pallares et al. The *x*-axis shows (symmetric) variance-covariance matrices for true effects in the high-sugar and control diets. The variance-covariance matrices displayed are the only matrices to which *mash* assigned non-zero weight (from a much larger set of possible covariance matrices, following a variable selection procedure). Variance-covariance matrices are scaled by a constant for each of interpretability. Abbreviations: C–control diet; HS–high-sugar diet.

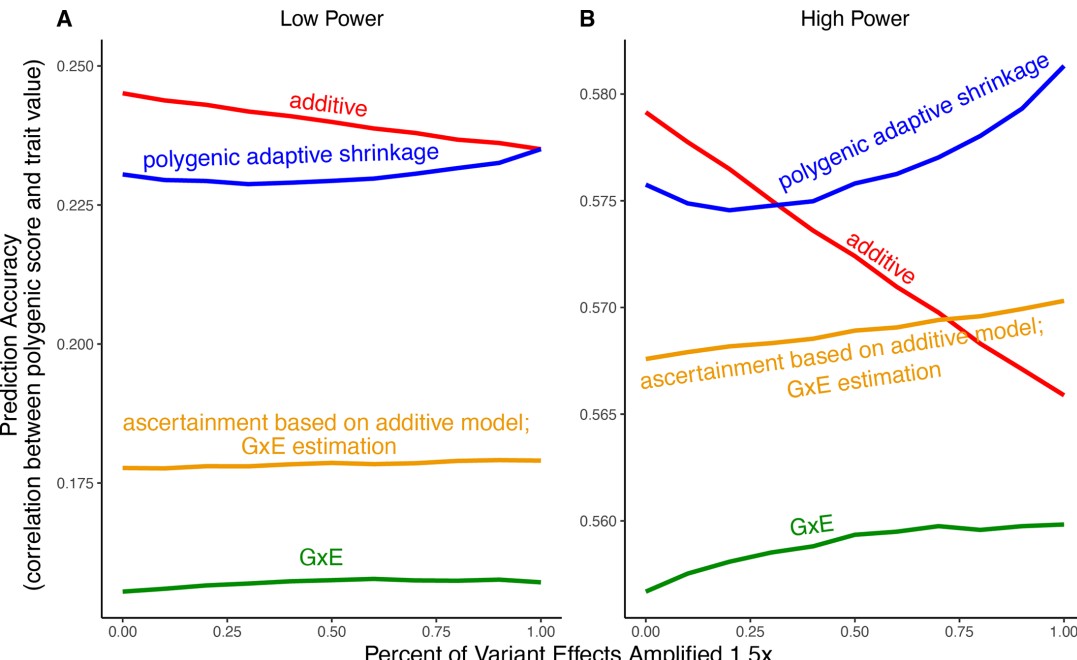

**Appendix 1—figure 5.** Polygenic score performance for context-aware models with greater amplification than in the main text. This figure uses the same simulation parameters as *Figure 4* of the main text, except that variant effects are amplified $1.5\times$ instead of $1.4\times$. In each simulation, a genome-wide association study (GWAS) is performed on 5000 biallelic variants, half of which have no effect in either context. Of the other half, some percent of the variants (indicated on the *x*-axis) had effects $1.5\times$ larger in one of contexts and the remaining SNPs

*Appendix 1—figure 5 continued*

had equal effects in both contexts. The broad sense heritability was set to $0.4$ in all simulations. The *y*-axis shows the average, over 1000 simulations, of the out-of-sample Pearson correlation between polygenic score and trait value. (**A**) Results with a GWAS sample size of 1000 individuals. (**B**) Results with a GWAS size of $50,000$ individuals. We note that unlike in *Figure 5*, for amplification percentages greater than or equal to 75%, the strategy using ascertainment based on the additive model with gene-by-environment interactions (GxE) effect estimation (orange) outperforms the strategy of using the additive model for both tasks (red).

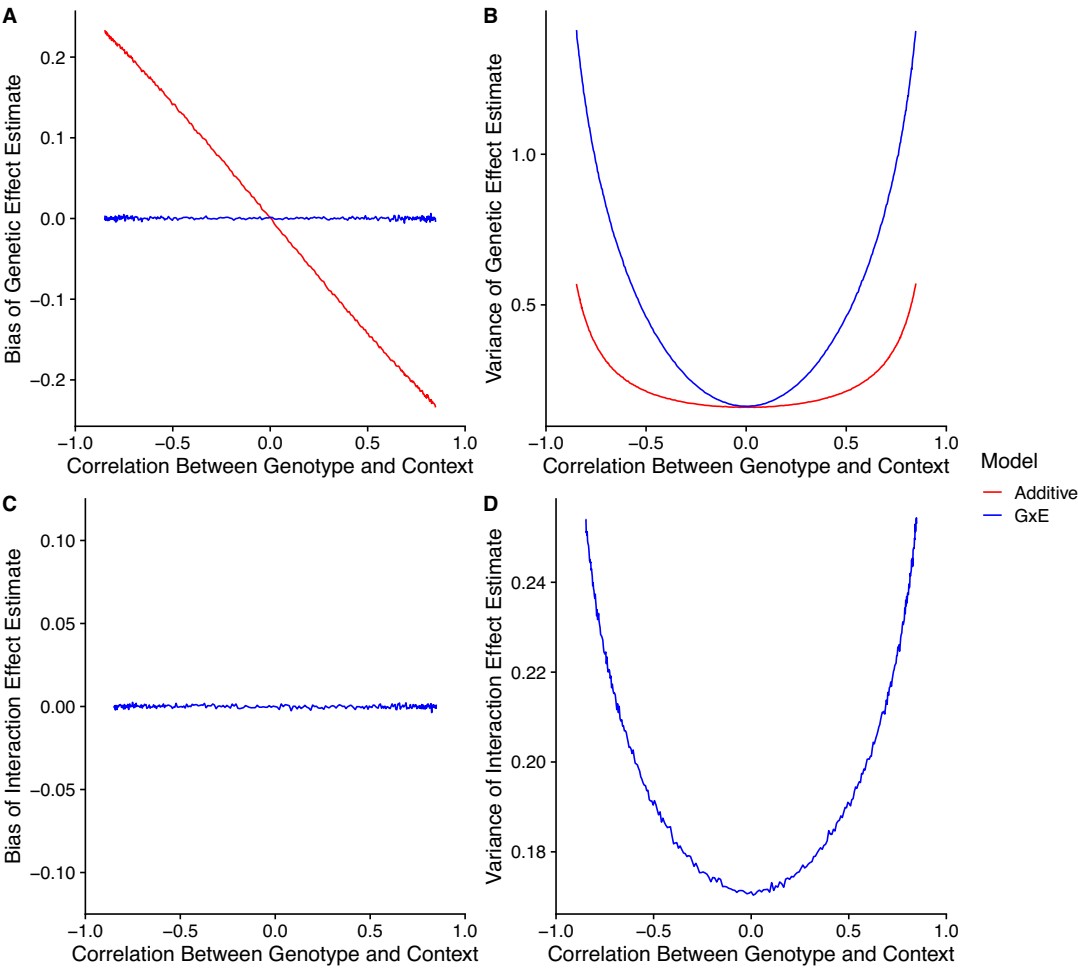

**Appendix 1—figure 6.** Model performance under correlation between genotype and context. (**A**) Bias of genetic effect estimate across different values of correlation. (**B**) Variance of genetic effect estimate across different values of correlation. (**C**) Bias of term estimating the interaction between genotype and context in an interaction model. (**D**) Variance of term estimating the interaction between genotype and context in an interaction model.

