## [Editor Report · eLife Assessment]

It is known from model organisms that genes' effects on traits are often modulated by environmental variables, but similar gene-by-environment (GxE) interactions have been difficult to detect using statistical analyses of genomic data, e.g., in humans. This study introduces a new framework to estimate gene-by-environment effects, treating it as a bias-variance tradeoff problem. The authors **convincingly** show that greater statistical power can be achieved in detecting GxE if an underlying model of polygenic GxE is assumed. This polygenic amplification model is a truly novel view with **fundamental** promise for the detection of GxE in genomic datasets, especially with continued development to detect more complex signals of amplification.

---

## [Referee Report · Reviewer #1 (Public review)]

Experiments in model organisms have revealed that the effects of genes on heritable traits are often mediated by environmental factors -- so-called gene-by-environment (or GxE) interactions. In human genetics, however, where indirect statistical approaches must be taken to detect GxE, limited evidence has been found for pervasive GxE interactions. The present manuscript argues that the failure of statistical methods to detect GxE may be due to how GxE is modelled (or not modelled) by these methods.

The authors show, via re-analysis of an existing dataset in *Drosophila*, that a polygenic 'amplification' model can parsimoniously explain patterns of differential genetic effects across environments. (Work from the same lab had previously shown that the amplification model is consistent with differential genetic effects across the sexes for a number of traits in humans.) The parsimony of the amplification model allows for powerful detection of GxE in scenarios in which it pertains, as the authors show via simulation.

Before the authors consider polygenic models of GxE, however, they present a very clear analysis of a related question around GxE: When one wants to estimate the effect of an individual allele in a particular environment, when is it better to stratify one's sample by environment (reducing sample size, and therefore increasing the variance of the estimator) versus using the entire sample (including individuals not in the environment of interest, and therefore biasing the estimator away from the true effect specific to the environment of interest)? Intuitively, the sample-size cost of stratification is worth paying if true allelic effects differ substantially between the environment of interest and other environments (i.e., GxE interactions are large), but not worth paying if effects are similar across environments. The authors quantify this trade-off in a way that is both mathematically precise and conveys the above intuition very clearly. They argue on its basis that, when allelic effects are small (as in highly polygenic traits), single-locus tests for GxE may be substantially underpowered.

The paper is an important further demonstration of the plausibility of the amplification model of GxE, which, given its parsimony, holds substantial promise for the detection and characterization of GxE in genomic datasets. However, the empirical and simulation examples considered in the paper (and previous work from the same lab) are somewhat "best-case" scenarios for the amplification model, with only two environments and with these environments amplifying equally the effects of only a single set of genes. It would be an important step forward to demonstrate the possibility of detecting amplification in more complex scenarios, with multiple environments each differentially modulating the effects of multiple sets of genes. This could be achieved via simulations similar to those presented in the current manuscript.

Comments on revisions:

The authors have (with reasonable justification) said that my main recommendations for strengthening the conclusions of the paper are beyond its scope, and they have thoughtfully responded to my (and the other reviewer's) other comments. The paper is now more clearly written---in particular, the connection between the single-locus bias-variance tradeoff calculations and the polygenic results is much more transparent than before. Given that the authors have (again, with fair justification) chosen not to address my major comment, my broad assessment of the paper is unchanged---I think it is an important contribution to a critical topic---and I have no further comments for its improvement (though I note an issue with figure referencing in the captions of Supplementary Figs S2 and S3).

---

## [Author Response]

The following is the authors’ response to the original reviews.

**Public Reviews:**

**Reviewer #1 (Public Review):**
Experiments in model organisms have revealed that the effects of genes on heritable traits are often mediated by environmental factors---so-called gene-by-environment (or GxE) interactions. In human genetics, however, where indirect statistical approaches must be taken to detect GxE, limited evidence has been found for pervasive GxE interactions. The present manuscript argues that the failure of statistical methods to detect GxE may be due to how GxE is modelled (or not modelled) by these methods.The authors show, via re-analysis of an existing dataset in *Drosophila*, that a polygenic ‘amplification’ model can parsimoniously explain patterns of differential genetic effects across environments. (Work from the same lab had previously shown that the amplification model is consistent with differential genetic effects across the sexes for several traits in humans.) The parsimony of the amplification model allows for powerful detection of GxE in scenarios in which it pertains, as the authors show via simulation.Before the authors consider polygenic models of GxE, however, they present a very clear analysis of a related question around GxE: When one wants to estimate the effect of an individual allele in a particular environment, when is it better to stratify one’s sample by environment (reducing sample size, and therefore increasing the variance of the estimator) versus using the entire sample (including individuals not in the environment of interest, and therefore biasing the estimator away from the true effect specific to the environment of interest)? Intuitively, the sample-size cost of stratification is worth paying if true allelic effects differ substantially between the environment of interest and other environments (i.e., GxE interactions are large), but not worth paying if effects are similar across environments. The authors quantify this trade-off in a way that is both mathematically precise and conveys the above intuition very clearly. They argue on its basis that, when allelic effects are small (as in highly polygenic traits), single-locus tests for GxE may be substantially underpowered.The paper is an important further demonstration of the plausibility of the amplification model of GxE, which, given its parsimony, holds substantial promise for the detection and characterization of GxE in genomic datasets. However, the empirical and simulation examples considered in the paper (and previous work from the same lab) are somewhat “best-case” scenarios for the amplification model, with only two environments, and with these environments amplifying equally the effects of only a single set of genes. It would be an important step forward to demonstrate the possibility of detecting amplification in more complex scenarios, with multiple environments each differentially modulating the effects of multiple sets of genes. This could be achieved via simulations similar to those presented in the current manuscript.
**Reviewer #2 (Public Review):**
Summary:Wine et al. describe a framework to view the estimation of gene-context interaction analysis through the lens of bias-variance tradeoff. They show that, depending on trait variance and context-specific effect sizes, effect estimates may be estimated more accurately in context-combined analysis rather than in context-specific analysis. They proceed by investigating, primarily via simulations, implications for the study or utilization of gene-context interaction, for testing and prediction, in traits with polygenic architecture. First, the authors describe an assessment of the identification of context-specificity (or context differences) focusing on “top hits” from association analyses. Next, they describe an assessment of polygenic scores (PGSs) that account for context-specific effect sizes, showing, in simulations, that often the PGSs that do not attempt to estimate context-specific effect sizes have superior prediction performance. An exception is a PGS approach that utilizes information across contexts. Strengths:The bias-variance tradeoff framing of GxE is useful, interesting, and rigorous. The PGS analysis under pervasive amplification is also interesting and demonstrates the bias-variance tradeoff.Weaknesses:The weakness of this paper is that the first part -- the bias-variance tradeoff analysis -- is not tightly connected to, i.e. not sufficiently informing, the later parts, that focus on polygenic architecture. For example, the analysis of “top hits” focuses on the question of testing, rather than estimation, and testing was not discussed within the bias-variance tradeoff framework. Similarly, while the PGS analysis does demonstrate (well) the bias-variance tradeoff, the reader is left to wonder whether a bias-variance deviation rule (discussed in the first part of the manuscript) should or could be utilized for PGS construction.

We thank the editors and the reviewers for their thoughtful critique and helpful suggestions throughout. In our revision, we focused on tightening the relationship between the analytical single variant bias-variance tradeoff derivation and the various empirical analyses that follow.

We improved discussion of our scope and what is beyond our scope. For example, our language was insufficiently clear if it suggested to the editor and reviewers that we are developing a method to characterize polygenic GxE. Developing a new method that does so (let alone evaluating performance across various scenarios) is beyond the scope of this manuscript.

Similarly, we clarify that we use amplification only as an example of a mode of GxE that is not adequately characterized by current approaches. We do not wish to argue it is an omnibus explanation for all GxE in complex traits. In many cases, a mixture of polygenic GxE relationships seems most fitting (as observed, for example, in Zhu et al., 2023, for GxSex in human physiology).

**Recommendations for the authors:**

**Reviewer #1 (Recommendations For The Authors):**
MAJOR COMMENTThe amplification model is based on an understanding of gene networks in which environmental variables concertedly alter the effects of clusters of genes, or modules, in the network (e.g., if an environmental variable alters the effect of some gene, it indirectly and proportionately alters the effects of genes downstream of that gene in the network---or upstream if the gene acts as a bottleneck in some pathway). It is clear in this model that (i) multiple environmental variables could amplify distinct modules, and (ii) a single environmental variable could itself amplify multiple separate modules, with a separate amplification factor for each module.However, perhaps inspired by their previous work on GxSex interactions in humans, the authors’ focus in the present manuscript is on cases where there are only two environments (“control” and “high-sugar diet” in the *Drosophila* dataset that they reanalyze, and “A” and “B” in their simulations [and single-locus mathematical analysis]), and they consider models where these environments amplify only a single set of genes, i.e., with a single amplification factor. While it is of course interesting that a single-amplification-factor model can generate data that resemble those in the *Drosophila* dataset that the authors re-analyze, most scenarios of amplification GxE will presumably be more complex. It seems that detecting amplification in these more complex scenarios using methods such as the authors do in their final section will be correspondingly more difficult. Indeed, in the limit of sufficiently many environmental variables amplifying sufficiently many modules, the scenario would resemble one of idiosyncratic single-locus GxE which, as the authors argue, is very difficult to detect. That more complex scenarios of amplification, with multiple environments separately amplifying multiple modules each, might be difficult to detect statistically is potentially an important limitation to the authors’ approach, and should be tested in their simulations.

We agree that characterizing GxE when there is a mixture of drivers of context-dependency is difficult. Developing a method that does so across multiple (and perhaps not pre-defined) contexts is of high interest to us but beyond the scope of the current manuscript

We note that for GxSex, modeling this mixture does generally improve phenotypic prediction, and more so in traits where we infer amplification as a major mode of GxE.

MINOR COMMENTSLines 88-90: “This estimation model is equivalent to a linear model with a term for the interaction between context and reference allele count, in the sense that context-specific allelic effect estimators have the same distributions in the two models.”Does this equivalence require the model with the interaction term also to have an interaction term for the intercept, i.e., the slope on a binary variable for context (since the generative model in Eq. 1 allows for context-specific intercepts)?

It does require an interaction term for the intercept. This is e_i (and its effect beta_E) in Eq. S2 (line 70 of the supplement).

Lines 94-96: Perhaps just a language thing, but in what sense does the estimation model described in lines 92-94 “assume” a particular distribution of trait values in the combined sample? It’s just an OLS regression, and one can analyze its expected coefficients with reference to the generative model in Eq. 1, or any other model. To say that it “assumes” something presupposes its purpose, which is not clear from its description in lines 92-94.

We corrected “assume” to “posit”.

Lines 115-116: It should perhaps be noted that the weights wA and wB need not sum to 1.

Indeed; it is now explicitly stated.

Lines 154-160: I think the role of r could be made even clearer by also discussing why, when VA>>VB, it is better to use the whole-sample estimate of betaA than the sample-A-specific estimate (since this is a more counterintuitive case than the case of VA<<VB discussed by the authors).

This is addressed in lines 153-154, stating: “Typically, this (VA<<VB) will also imply that the additive estimator is greatly preferable for estimating β_B , as β_B will be extremely noisy”

Line 243 and Figure 4 caption: The text states that the simulated effects in the high-sugar environment are 1.1x greater than those in the control environment, while the caption states that they are 1.4x greater.

We have corrected the text to be consistent with our simulations.

TYPOS/WORDINGLine 14: “harder to interpret”  “harder-to-interpret”Line 22: We  weLine 40: “as average effect” -> “as the average effect”?Line 57: “context specific”  “context-specific”Line 139: “re-parmaterization”  “re-parameterization”Lines 140, 158, 412: “signal to noise”  “signal-to-noise”Figure 3C,D: “pule rate”  “pulse rate”The caption of Figure 3: “conutinous”  “continuous”Line 227: “a variant may fall”  “a variant may fall into”Line 295: “conferring to more GxE”  “conferring more GxE” or “corresponding to more GxE”? This is very pedantic, but I think “bias-variance” should be “bias--variance” throughout, i.e., with an en-dash rather than a hyphen.

We have corrected all of the above typos.

**Reviewer #2 (Recommendations For The Authors):**
(This section repeats some of what I wrote earlier).- First polygenic architecture part: the manuscript focuses on “top hits” in trying to identify sets of variants that are context-specific. This “top hits” approach seems somewhat esoteric and, as written, not connected tightly enough to the bias-variance tradeoff issue. The first section of the paper which focuses on bias-variance trade-off mostly deals with estimation. The “top hits” section deals with testing, which introduces additional issues that are due to thresholding. Perhaps the authors can think of ways to make the connection stronger between the bias-variance tradeoff part to the “top hits” part, e.g., by introducing testing earlier on and/or discussion estimation in addition to testing in the “top hits” part of the manuscript. The second polygenic architecture part: polygenic scores that account for interaction terms. Here the authors focused (well, also here) on pervasive amplification in simulations. This part combines estimation and testing (both the choice of variants and their estimated effects are important). In pervasive amplification the idea is that causal variants are shared, the results may be different than in a model with context-specific effects and variant selection may have a large impact. Still, I think that these simulations demonstrate the idea developed in the bias-variance tradeoff part of the paper, though the reader is left to wonder whether a bias-variance decision rule should or could be utilized for PGS construction.

In both of these sections we discuss how the consideration of polygenic GxE patterns alters the conclusions based on the single-variant tradeoff. In the “top hits” section, we show that single-variant classification itself, based on a series of marginal hypothesis tests alone, can be misleading. The PGS prediction accuracy analysis shows that both approaches are beaten by the polygenic GxE estimation approach. Intuitively, this is because the consideration of polygenic GxE can mitigate both the bias and variance, as it leverages signals from many variants.

We agree that the links between these sections of the paper were not sufficiently clear, and have added signposting to help clarify them (lines 176-180; lines 275-277; lines 316-321).

- Simulation of GxDiet effects on longevity: the methods of the simulation are strange, or communicated unclearly. The authors’ report (page 17) poses a joint distribution of genetic effects (line 439), but then, they simulated effect estimates standard errors by sampling from summary statistics (line 445) rather than simulated data and then estimating effect and effect SE. Why pose a true underlying multivariate distribution if it isn’t used?

We rewrote the Methods section “Simulation of GxDiet effects on longevity in *Drosophila* to make our simulation approach clearer (lines 427-449). We are indeed simulating the true effects from the joint distribution proposed. However, in order to mimic the noisiness of the experiment in our simulations, we sample estimated effects from the true simulated effects, with estimation noise conferring to that estimated in the Pallares et al. dataset (i.e., sampling estimation variances from the squares of empirical SEs).

- How were the “most significantly associated variants” selected into the PGS in the polygenic prediction part? Based on a context-specific test? A combined-context test of effect size estimates?

For the “Additive” and “Additive ascertainment, GxE estimation” models (red and orange in Fig. 5, respectively), we ascertain the combined-context set. For the “GxE” and “polygenic GxE” (green and blue in Fig. 5, respectively) models, we ascertain in a context-specific test. We now state this explicitly in lines 280-288 and lines 507-526.

- As stated, I find the conclusion statement not specific enough in light of the rest of the manuscript. “the consideration of polygenic GxE trends is key” - this is very vague. What does it mean “to consider polygenic GxE trends” in the context of this paper? I can’t tell. “The notion that complex trait analyses should combine observations at top associated loci” - I don’t think the authors really refer to combining “observations”, rather perhaps combine information from top associated loci. But this does not represent the “top hits” approach that merely counts loci by their testing patterns. “It may be a similarly important missing piece...” What does “it” refer to? The top loci? What makes it an important missing piece?

We rewrote the conclusion paragraph to address these concerns (lines 316-321).